# Survey on Coverage Path Planning with Unmanned Aerial Vehicles

**Tauã M. Cabreira \*,† [ID], Lisane B. Brisolara † [ID] and Ferreira Paulo R. Jr. † [ID]**

Programa de Pós-Graduação em Computação (PPGC), Universidade Federal de Pelotas (UFPel),
96010-610 Pelotas, Brazil; lisane@inf.ufpel.edu.br (L.B.B.); paulo@inf.ufpel.edu.br (P.R.F.J.)
\*   Correspondence: tmcabreira@inf.ufpel.edu.br; Tel.: +55-53-3284-3890
†   These authors contributed equally to this work.

**Abstract:** Coverage path planning consists of finding the route which covers every point of a certain area of interest. In recent times, Unmanned Aerial Vehicles (UAVs) have been employed in several application domains involving terrain coverage, such as surveillance, smart farming, photogrammetry, disaster management, civil security, and wildfire tracking, among others. This paper aims to explore and analyze the existing studies in the literature related to the different approaches employed in coverage path planning problems, especially those using UAVs. We address simple geometric flight patterns and more complex grid-based solutions considering full and partial information about the area of interest. The surveyed coverage approaches are classified according to a classical taxonomy, such as no decomposition, exact cellular decomposition, and approximate cellular decomposition. This review also contemplates different shapes of the area of interest, such as rectangular, concave and convex polygons. The performance metrics usually applied to evaluate the success of the coverage missions are also presented.

**Keywords:** unmanned aerial vehicles; coverage path planning; terrain coverage; exact cellular decomposition; approximate cellular decomposition

---

## 1. Introduction

Unmanned Aerial Vehicles (UAVs) have increasingly been used in a wide range of applications, such as surveillance [1], smart farming [2], photogrammetry [3], disaster management, civil security [4], wildfire tracking [5], cloud monitoring [6], structure supervision [7], and power line inspection [8]. The UAVs consist of aerial platforms with no pilots on-board the vehicle. Such platforms are remotely and manually operated by a human, but they also perform automated pre-programmed flights. Autonomous flights can be executed using intelligent systems integrated with on-board sensors.

The Coverage Path Planning (CPP) problem is classified as a motion planning subtopic in robotics, where it is necessary to build a path for a robot to explore every location in a given scenario [9]. Despite the technological progress in this type of aerial platform regarding autonomous flight, it is important to emphasize that the phases concerning take-off, mission execution, and landing are usually assisted by two people for each UAV due to safety measures. The pilot supervises the mission and may change the flight mode to manual in case of a failure or an emergency during the flight, while the base operator monitors the navigation data during the mission execution, such as altitude variation and battery discharge [10].

### 1.1. UAV Classification

The UAVs can be classified into two main top-level configurations, fixed-wing and rotary-wing. Both types present specific advantages and challenges considering the control and guidance system [11].

The fixed-wing UAV presents rigid wings with an airfoil which allows flying based on the lift created by the forward airspeed. The navigation control is obtained using control surfaces in the wings (aileron, elevator, and rudder). The aerodynamics support longer endurance flights and loitering, also allowing high-speed motion. Besides, these vehicles can carry heavier payloads in comparison with rotary-wing vehicles. However, these platforms need a runway to take-off and land and are not able to perform hovering tasks, since they need to constantly move during missions [12].

The rotary-wing presents maneuverability advantages using rotary blades. These platforms are able to perform vertical take-off and landing (VTOL), low-altitude flights, and hovering tasks. The use of rotary blades produces aerodynamic thrust forces and does not require relative velocity [11]. This type of aerial platform can also be classified into single-rotor (helicopter) and multi-rotor (quadcopter and hexacopter).

The single-rotor has two rotors, the main one for navigation and the tail one for controlling the heading. These vehicles are usually able to vertically take-off and land and they do not require airflow over the blades in order to move forward. Instead, the blades themselves create the needed airflow. A gas motor enables even longer endurance flights in comparison with multi-rotors. This type of vehicle can carry high payloads, such as sensors and manipulators while performing hovering tasks and long-time flights in outdoors missions, However, these platforms present mechanical complexity and elevated cost [12].

The multi-rotor can be divided into subclasses regarding the number of rotor blades. The most common are the quadcopter and the hexacopter, but tricopters and octocopters have also been developed. Multi-rotors are fast and agile platforms and are able to perform demanding maneuvers. They are also capable of hovering or moving along a target. Nevertheless, these platforms have limited payload and endurance. Mechanical and electrical complexity is quite low as these parts are abstracted away within the flight and motor controllers [12]

There is also the hybrid UAV, which is a specific type of aerial platform including the advantages of both, fixed-wing and rotary-wing, thus having the capability of VTOL, high-flight speed and increased flight time. These vehicles can be classified into Convertiplanes and Tail-Sitters. The former one consists of a hybrid platform that performs the basic maneuvers keeping the aircraft reference line in the horizontal direction. The latter one is a platform able to vertically take off and land on its tail, tilting forward in order to achieve horizontal flight [13]. Finally, other types of classifications related to UAVs may be found in the literature considering mission requirements, such as altitude and endurance. In these cases, the aerial platforms can be categorized considering low, medium, and high altitude, and also considering short and long endurance [14].

*1.2. Overview of the Existing Surveys*

A wide range of surveys presenting studies related to control, perception, and guidance of UAVs is addressed in the literature, such as system identification approaches for low-cost UAVs [15], trajectory planning with and without differential constraints through an environment with obstacles [16], UAV autonomous guidance under uncertainty conditions [17], helicopter navigation and control techniques [11], and perception and state estimation for UAVs [12]. Considering specific applications, Kanistras et al. [18] presents a survey exploring studies of UAVs employed in traffic monitoring and management, while Colomina and Molina [19] addresses the UAV technology for precision agriculture. A review on landing techniques is presented by Gautam et al. [20], providing a wide outlook on the controller design.

Choset [9] presents a survey on CPP for mobile robots, where the author classifies the approaches either as heuristic or complete. In the heuristic approaches, the robots follow a set of simple rules defining their behavior, but such methods do not present a guarantee for coverage success. On the other hand, complete methods can provide these guarantees using the cellular decomposition of the environment, which consists of space discretization into cells to simplify the coverage in each sub-region. Another important issue mentioned by the author is flight time, which can be minimized

using multiple robots and reducing the number of turning maneuvers. Finally, the author highlights the available environment information. Several approaches admit previous knowledge of the robot regarding the search area (offline), while sensor-based approaches acquire such information in real-time during the coverage (online).

The most recent survey regarding CPP presents several approaches and techniques to perform mostly missions with land vehicles [21]. Considering exploration under uncertainty, Juliá et al. [22] present a study about unknown environment mapping strategies. The robots are supposed to autonomously explore the environment to collect data and build a navigation map. With the absence of global positioning information, it is necessary to constantly correct the robot positioning and orientation estimation using simultaneous localization and mapping techniques. Multiple robots can be employed to either reduce exploration time or improve map quality, but require coordination strategies. In such strategies, the robots may share perceptions and construct a common map of the workspace. This global map may be built either as a centralized or distributed way.

*1.3. Motivation of This Review*

The existing surveys related to UAVs address important issues, such as control, perception, and guidance. A few surveys address the CPP problem, but only considering land vehicles and briefly mentioning the UAVs as an extension of these vehicles. Although land exploration techniques revised in the previous surveys can be extended and applied to UAVs, several additional aspects must be considered when dealing with aerial vehicles such as vehicle's physical characteristics, endurance, maneuverability limitations, restricted payload, environmental external conditions, among others. On-board cameras and sensors can increase the vehicle's weight and reduce endurance, which is quite limited especially in multi-rotors. In such vehicles, endurance is about 20–25 min, even in more sophisticated models released in 2018 [23]. Moreover, turning maneuvers [24,25] and wind fields [26] increase energy consumption in outdoor missions.

This paper presents a survey on coverage path planning. Our review considers only approaches related to unmanned aerial vehicles. The classic taxonomy defined by Choset [9] was adopted to classify the existing approaches according to the cellular decomposition technique employed. Approaches with no decomposition and methods using exact and approximate cellular decomposition are considered. The latter ones, also known as grid-based methods, are divided into two subsections, full and partial information. The full information subsection explores algorithms which guarantee the completeness of the mission covering all the decomposed cells, while the partial information subsection presents bio-inspired methods performing coverage under uncertainties. This review considers different shapes of the area of interest, including rectangular, concave, and convex polygons. These scenarios are also categorized according to the available information to perform coverage. Moreover, we explore performance metrics usually applied to evaluate the success of the coverage missions.

This survey is organized as follows: Section 2 addresses coverage path planning problem, describing how the areas of interest are characterized and how they are treated in the flight planning. The different decomposition techniques employed to split and discretize the areas of interest are presented as well as the performance metrics. Section 3 explores the simple flight patterns adopted in areas of interest with no decomposition technique. Section 4 addresses coverage solutions for areas of interest discretized using the exact cellular decomposition. Section 5 presents coverage approaches for areas of interest discretized into a grid using approximate cellular decomposition. Section 6 summarizes the overall analysis, highlighting the main pros and cons of the revised CPP methods. Section 7 concludes the survey presenting possible gaps to explored in the future regarding coverage path planning using UAVs.

## 2. Coverage Path Planning

Given an area of interest composed by the robot's free space and its boundaries, the CPP problem consists of planning a path which covers the entire target environment considering the vehicle's motion

restrictions and sensor's characteristics, while avoiding passing over obstacles. In an aerial context, the workspace obstacles can represent no-flight zones (NFZ) that the UAV should not consider during the planning phase, e.g., areas next to airports or irrelevant buildings.

The target environment is usually split into non-intersecting regions called cells using a decomposition technique. The size and resolution of the cells may change according to the type of decomposition and a specific strategy should be applied in order to guarantee the complete coverage. For larger cells, several motions are necessary to fully cover only one unit, while in smaller cells a single motion is enough. These cells typically have the same size of a robot (terrestrial coverage) or are proportional to the sensor's range (aerial coverage), representing only one point in the projected path. The CPP problem is further explored in the next subsections consisting of the area of interest definition, the cellular decomposition techniques, the performance metrics, and the information availability.

### 2.1. Area of Interest

The area of interest can be represented by a sequence of $p$ vertices $\{v_1, ..., v_p\}$. Each vertex $v_i$ can be described by a pair of coordinates $(v_x(i), v_y(i))$, while its internal angle can be referred by $\gamma_i$. Considering $v_i$, the following vertex of the polygon can be described as $v_{next(i)}$, where $next(i) = i(\mathrm{mod}\ p) + 1$. An edge located between two vertices $v_i$ and $v_{next(i)}$ can be referred as $e_i$, while its length by $l_i = ||v_i - v_{next(i)}||$. Furthermore, the area may contain internal NFZ depicted as a sequence of obstacle-points $\{u_1, ..., u_p\}$. Figure 1 shows three examples of an area.

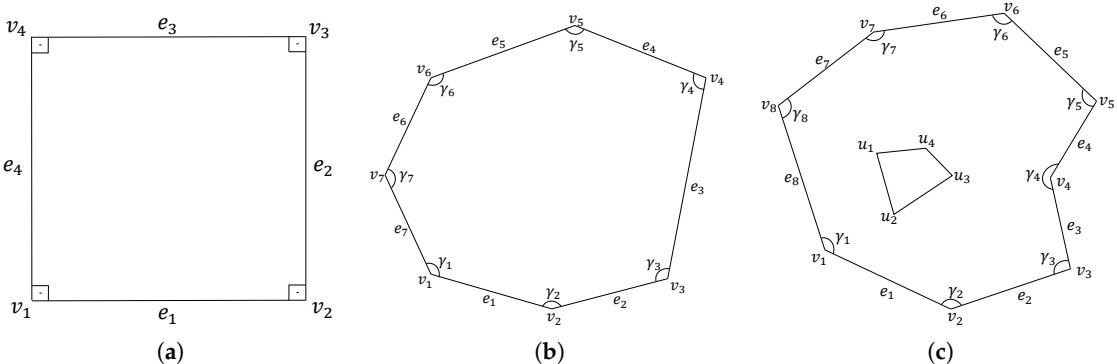

**Figure 1.** Different areas of interest explored during CPP missions: (**a**) Rectangular; (**b**) Convex Polygon; (**c**) Concave Polygon with No-Fly Zones.

The shape of the area of interest is a relevant factor to worry about during the coverage path planning. Some approaches explore only rectangular areas or simplify the area shape to a rectangle, while other ones support more complex shapes such as concave and convex polygons representing irregular areas. Some methods can even deal with areas of interest containing NFZ which must be avoided during coverage. These no-fly zones can represent regions where coverage is simply unnecessary or locations where the UAVs are not allowed to fly. Different decomposition techniques are usually adopted to reduce the concavities of complex areas or to split the area into smaller cells to facilitate the coverage task.

### 2.2. Cellular Decomposition

One of the major concerns about the CPP problem is to guarantee a complete coverage of the scenario. This is usually achieved applying cellular decomposition in the area of interest, splitting the target free space into cells in order to simplify the coverage [9]. In literature, there are different cellular decomposition methods and the most common used in CPP problem involving UAVs are exact and approximate cellular decomposition.

Exact cellular decomposition consists of splitting the workspace into sub-areas, also known as cells, whose re-union exactly occupy the target area. These cells are usually explored by simple motions as back-and-forth. In this way, the CPP problem can be reduced to motion planning from one cell to another [9]. These motions are performed between adjacent cells sharing a mutual border. Considering the adjacency graph representation, nodes can denote cells, while edges can identify neighbor cells, as depicted in Figure 2. Thus, the decomposed cells are created by sweeping a line from one side to another in the area of interest. The limits of the cells are defined by events triggered every time the sweep line crosses an obstacle boundary. The resulting decomposition can be stored as an adjacency graph and a search can be executed in order to find a connected path exploring each node only once. The final coverage path is composed of the simple motions performed inside the cells and the inter-cell connections [21].

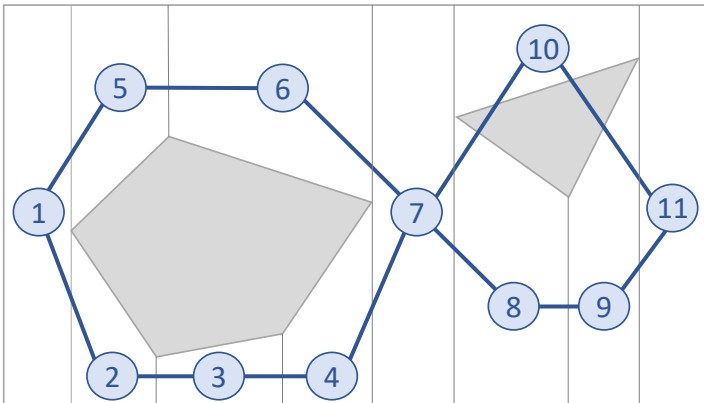

**Figure 2.** Adjacency graph representing the workspace splitted into cells.

There are two important exact cellular decomposition techniques worth to be mentioned: trapezoidal decomposition and boustrophedon decomposition (Boustrophedon literally refers to "the way of the ox", an analogy to the animal's motion while dragging a plow in a field.), as shown in Figure 3a,b, respectively. The former one divides the area of interest into convex trapezoidal cells, performs back-and-forth motions and uses an exhaustive walk to determine the cells exploration sequence to fulfill the coverage. The latter one creates non-convex larger cells considering only obstacle-vertices. A sweeping line is prolonged in both ways of the obstacles and these zones are called critical points. The boustrophedon decomposition is able to diminish the amount of trapezoidal cells and minimize the coverage path length in comparison to the trapezoidal decomposition.

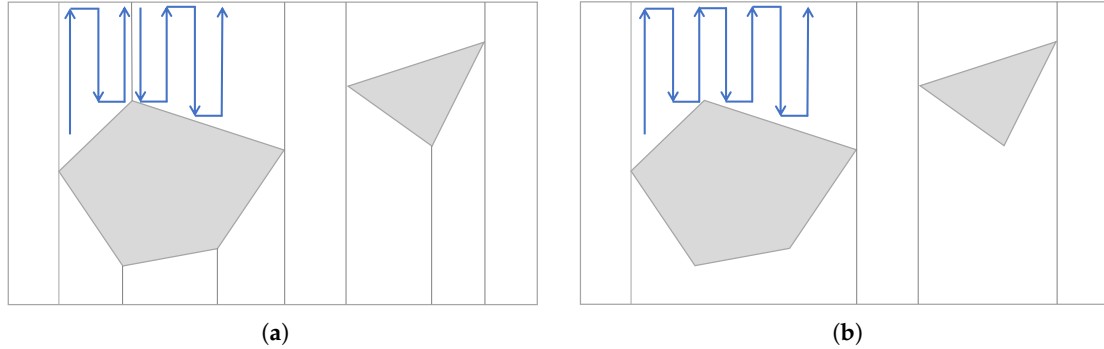

|         |         |
| :-----: | :-----: |
| (**a**) | (**b**) |

**Figure 3.** Two types of exact cellular decomposition: (**a**) Trapezoidal decomposition; (**b**) Boustrophedon decomposition.

The approximate cellular decomposition technique discretizes the area into a set of regular cells [9]. These regular cells usually assume a square form, but they can also be represented either in a triangle

or hexagonal form. Grid-based methods can be applied over approximate areas to generate coverage paths [21]. The size of the cells usually fits the robots dimensions when considering coverage using land robots. However, in aerial coverage the UAVs fly at a certain altitude from the ground carrying a camera as a sensor to perform the task. In this case, the size of the cells is proportional to the footprint of the camera in the UAV, as illustrated in Figure 4a, and the grid resolution is obtained through the image requirements, such as resolution and overlapping rates, and the image sensor characteristics.

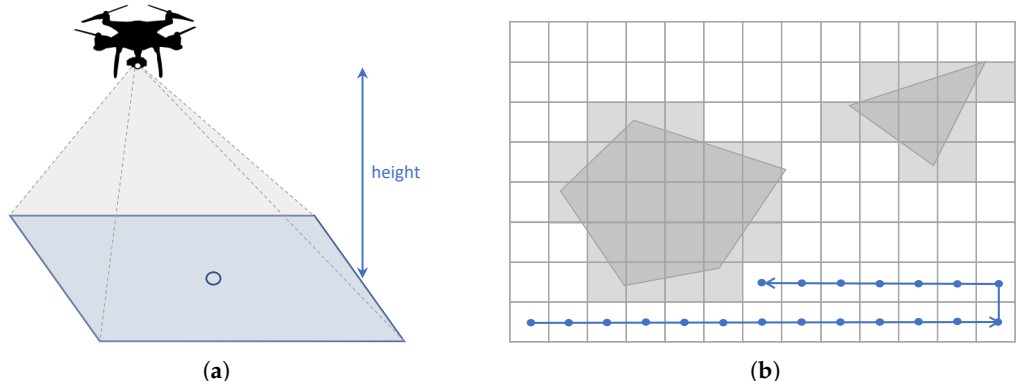

**Figure 4.** Approximate cellular decomposition: (**a**) Projected area; (**b**) Regular grid with waypoints.

The UAV coverage path is composed of a set of $k$ waypoints $\{w_1, ..., w_k\}$. Each waypoint $w_i$ represents a navigation command to the vehicle, such as take-off, change of speed or move to a specific location, and contains information about the latitude, longitude and altitude. Since the waypoints have all the necessary localization information to guide the vehicle and the cells are proportional to the footprint of the camera, we can simplify the problem assuming that the center of each cell refers to a waypoint, as shown in Figure 4b.

*2.3. Performance Metrics*

Coverage algorithms must consider several issues to guarantee the success of a coverage mission, such as the complexity of the area of interest, the presence or not of no-fly zones and the possibility to employ cellular decomposition techniques. Furthermore, the coverage algorithms should generate coverage paths according to the application requirements. For instance, the main goal of a photogrammetric sensing application is to create an orthomosaic composed by a set of overlapping aerial photographs. In this case, an application requirement is to guarantee the necessary quantity of frontal and side superposition in the pictures. Another necessary requirement for this type of application is the resolution, which can be calculated as ground sampling distance (GSD) [27]. The GSD is the length on the ground corresponding to the side of one pixel in the image, or the distance between pixel centers measured on the ground. The lower the flight altitude of the UAV, smaller the GSD and the better the image quality.

For this reason, the performance metrics used to evaluate the candidate solutions for a coverage path must fulfill the application requirements. In addition, it should take into account whether the coverage is simple or continuous. In a simple coverage, the area of interest is covered only once, while in a continuous coverage the scenario is swept several times. In both cases, the coverage can be performed by a single or multiple vehicles. Specific metrics for continuous coverage include the number of detected objects/events, interval and frequency of visits in each environment cell, the quadratic mean of intervals (QMI) [28], and the standard deviation of frequencies (SDF) [28].

Considering the context of a simple coverage, the most common performance metrics found in the literature are: the total travelled distance or the path length [29,30], the time-to-complete a mission [31,32], the area coverage maximization [33], and the number of turning maneuvers [34,35]. Minimizing the coverage path length lies in a trade-off with the area coverage maximization. In a workspace split by cellular decomposition, the path length should not be only minimized inside each

cell, but also in the intermediate routes between adjacent cells, i.e., the path connecting the end of one cell and the start of the next cell. Keeping the UAV inside the area of interest, avoiding flying over locations previously visited and flying at higher altitudes also minimize the coverage distance.

The use of multiple robots reduces the coverage flight time. For one robot, however, one can consider the area covered per unit path length travelled. Minimizing this quantity improves time-to-completion for both single and multi-robot coverage [9]. The use of multiple robots usually requires a coordination process that includes splitting the area of interest and assigning the resulting sub-areas among the UAVs. The workspace can be divided and assigned in two different steps [34] or simultaneously using a negotiation protocol through a distributed way [10] considering relative capabilities of the vehicles. However, a full solution including optimal area decomposition, allocation and efficient coverage is a cooperative control problem and usually is classified as NP-hard. Besides that, once a vehicle is out of the mission or the scenario changes, a reconfiguration process is necessary to divide and assign the areas to the remaining vehicles. Therefore, many studies simplify this problem considering that the vehicles fly at distinct altitudes in order to avoid collisions.

Lastly but not least, there is the number of turning maneuvers often employed as the main performance metric in coverage. When an aerial vehicle executes a turning maneuver, it should reduce its speed, rotate and increase its speed again. Thus, the greater the number of executed maneuvers, the greater the time and the energy spent. In this way, the minimization of the number of maneuvers is often explored by the authors in order to indirectly save energy and prolong the mission time. The authors often connect metrics such as path length, time-to-complete a mission and number of turns with energy consumption trying to minimize them in order to save energy. However, for an efficient energy saving regarding UAVs, further features need to be investigated as vehicle's motions and constraints, turning angles, and optimal speeds. As stated by Di Franco and Buttazzo [25], different distances may have different optimal speeds with minimum energy consumption depending on their length. Therefore, as the major technological boundary using UAVs, the energetic consumption has attracted the interest of researchers [24,31–33,36] and has become the main optimization criteria due to the limited endurance of UAVs in coverage path planning missions.

## 2.4. Information Availability

The type of solution adopted for a coverage mission with UAVs depends on the amount of information available about the workspace. Assuming a dynamic context where the information may constantly change or is not fully available, one may consider a randomly decision-making. To deal with this scenario, the vehicle must employ on-board sensors to gather workspace data to perform the coverage, interleaving between the planning and execution of the path. This type of online coverage is sometimes called sensor-based coverage as it uses sensor information to drive the coverage operation. The vehicle does not have the complete knowledge (or full information) about the workspace at the beginning and should re-construct a full map in order to successfully execute the mission. The challenge here is to keep updated data while dealing with dynamic behavior, e.g., the localization of a moving target.

On the other hand, some coverage approaches have all the information available and are aware of the scenario's layout before the planning phase. In this case, the coverage is offline and is usually sectioned into three sequential main steps: decomposition, planning and execution. First, a cellular decomposition technique is applied over the area in order to discretized and split the workspace. Second, a coverage path planning algorithm having full knowledge about the environment searches for a solution according to the predefined performance metrics. Finally, the resulting path is executed and the mission is completed. It is important to highlight that during this final step, there is no external interference that may lead to a change in the established path, only exceptional cases such as pre-programmed failsafe.

## 3. No Decomposition

Coverage missions performed over regular-shaped and non-complex areas of interest with a single UAV usually do not require any type of decomposition. Simple geometric patterns are sufficient to explore such areas. The most common patterns are the back-and-forth (BF) and the spiral (SP). The former one is adopted by the Mission Planner, the most popular flight-control software [37], to enable the area coverage using a standard pattern. In this pattern, the motions consist of straight lines crossed in both directions with closed-angle maneuvers at the end of each round. The latter one usually performs motions passing by the external vertices of the area and reducing the radius towards the central point.

Some approaches explore only rectangular areas, according to Andersen [38], where the author compares different types of flight patterns. In this work, the back-and-forth pattern is classified into parallel and creeping line, as illustrated in Figure 5a,b, and it is preferable when the search area is large and there is no information about the likely target meeting point. The square flight pattern consists of straight lines and 90° turning maneuvers to the right side. The pattern starts at the central point and extends towards the borders, following a pattern similar to an ellipse shape, and it is normally employed when a uniform area coverage is desired, as shown in Figure 5c.

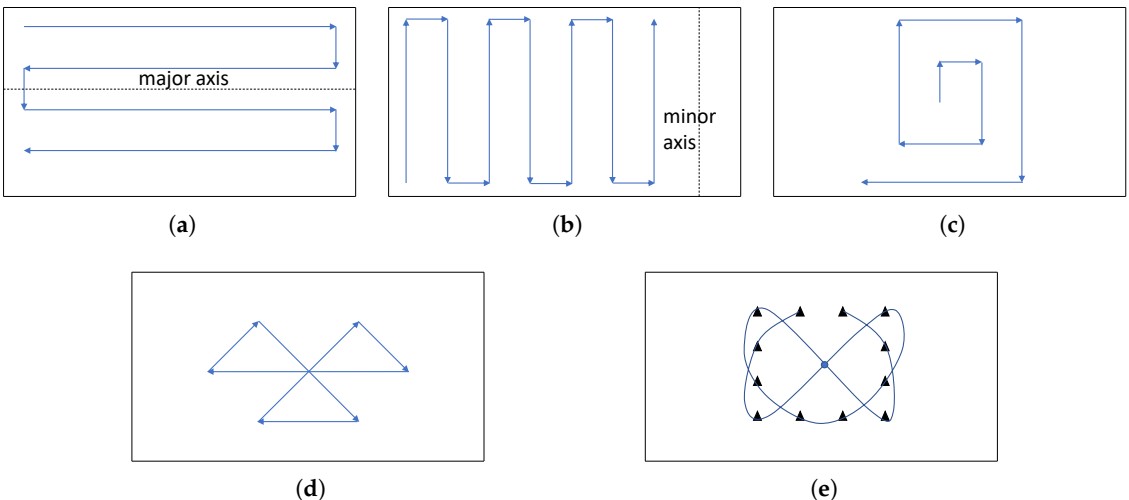

**Figure 5.** Simple flight patterns in rectangular areas with no decomposition: (**a**) Parallel; (**b**) Creeping Line; (**c**) Square; (**d**) Sector Search; (**e**) Barrier Patrol.

The sector search pattern, presented in Figure 5d, consists of a straight line with 120° turning maneuvers to the right when the vehicle reaches the border of the area. After three sectors, the path returns to the initial point at the center of the area. Then, the same pattern is repeated with 30° of displacement. The barrier patrol consists of the definition of 12 points spatially distributed in the search area, as illustrated in Figure 5e. The vehicle initiates its trajectory in the starting point and using a circular movement achieves the next point. From this point, instead of continuing the circular trajectory, it follows to the point closer to the right-corner and achieves the center point.

An analysis of the effect of wind disturbances in the mission execution time of BF coverage paths performed by a fixed-wing UAV is presented by Coombes et al. [39]. Using a circular area of interest covered with BF motions, the authors explore different sweep directions varying from 0 to 360 degrees in increments of 10 degrees with a predefined wind direction with six different speeds. According to the simulated experiments, the coverage direction must be perpendicular to the wind direction to minimize flight time. However, the turning maneuvers are directly affected by the choice of the perpendicular direction (clockwise or counterclockwise). The authors believe that in more complex scenarios decomposed into cells, the transition distance between those cells has more impact in the flight time than the wind direction.

Recent studies present energy-aware solutions exploring the dynamics and the behavior of the UAVs to save energy. Considering regular-shaped areas as rectangles and convex polygons, Di Franco and Buttazzo [25] presented an energy-aware back-and-forth CPP approach (E-BF) for photogrammetry with energy and resolution constraints imposed by the mission. In this approach, an algorithm determines the best configuration of back-and-forth motions at maximum altitude according to resolution constraints while minimizing the number of turns. The authors claim that it is possible to minimize the amount of energy flying at an optimal speed. This optimal speed varies according to the travelled distance.

The algorithm finds the first vertex of the longest edge and computes the scan direction parallel to it. Then, it calculates the number of stripes and waypoints, the distance between the stripes and consecutive waypoints, and the overlapping rates. Finally, a straight line connects the farthest vertex to the initial vertex. An algorithm improvement is also presented to avoid previously explored covered zones, as shown in Figure 6a. Turning the number of stripes in even and increasing the overlapping rate, the returning path can also be used as a scanning path, as illustrated in Figure 6b. The authors also proposed offline and online failsafe measures. The former one is checked offline in order to verify if the battery has enough energy to execute the mission. The latter one is checked online during the flight and constantly analyzes if the remaining energy is capable of bringing back the aerial vehicle to the starting point.

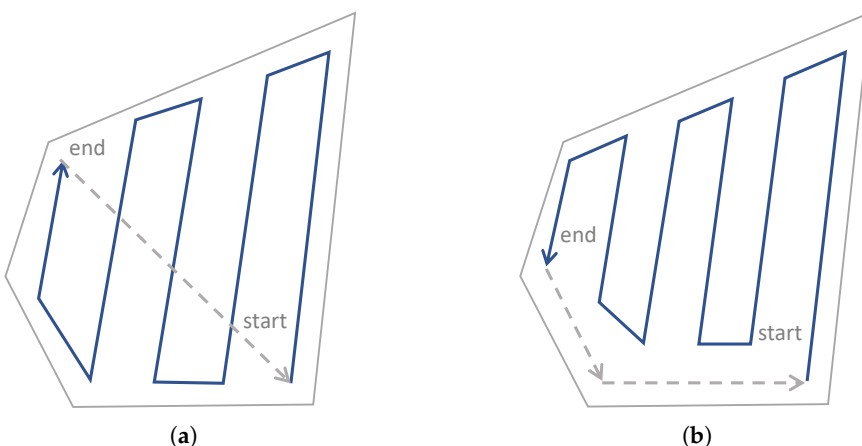

(**a**)　　　　　　　　　　　　　　　　　　　　　　　(**b**)

**Figure 6.** Energy-aware back-and-forth coverage path planning algorithm: (**a**) Odd number of stripes; (**b**) Even number of stripes.

An energy-aware spiral CPP algorithm (E-Spiral) is proposed by Cabreira et al. [40] for regular-shaped areas of interest. The algorithm consists of building a coverage path passing by each vertex of the area. Once the first coverage layer is completed, the algorithm should reduce the radius in order to move the vehicle towards the central point, as illustrated in Figure 7. The algorithm performs turning maneuvers with wider angles and does not need to reduce the speed to zero on every turn, which decreases the acceleration and deceleration periods. This behavior keeps the optimal speed adopted in straight segments of the path for longer periods, providing an even more effective energy saving than the one proposed by Di Franco and Buttazzo [25].

The E-BF [25] and the E-Spiral [40] adopt the energy model proposed by Di Franco and Buttazzo [24] derived from real measurements. The approaches are compared in simulations performed in 3750 different convex polygonal areas varying features, such as the angular points, the nonuniformity, and the size. Using a quadrotor, the authors also performed real experiments using both patterns in a rectangular and a polygonal area to analyze the energy spent during the missions. The E-Spiral overcomes the E-BF, both simulations and real flights, and can be considered the most efficient CPP approach for convex polygonal areas considering energy spent during missions.

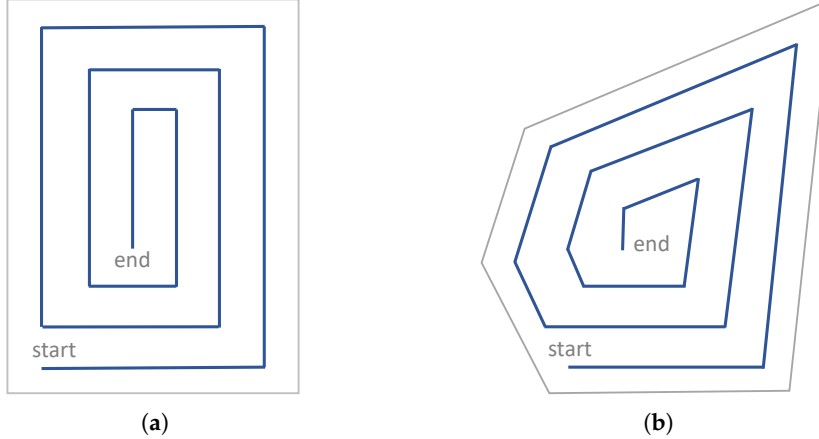

(**a**)                    (**b**)

**Figure 7.** Energy-aware spiral algorithm with energy and resolution constraints: (**a**) Rectangular area; (**b**) Polygonal area.

A triple-stage CPP algorithm for UAVs is presented in Li et al. [33], where the authors explore important features not addressed by Di Franco and Buttazzo [25], such as payload and power variation. The first step is to build a 3D terrain model using control points in order to obtain an analytical model. Next, the stable power consumption is calculated considering take-off weight, flight speed, and air friction. The authors consider that the vehicle moves with constant speed in a steady state, deriving the optimal speed aimed at minimizing energy. Furthermore, an energy consumption map is built to show the amount of energy spent on every part of the path. Finally, an optimization is performed with a Genetic Algorithm in order to discover minimum-cost paths comprising all vertices.

Energy-aware algorithms for smoothing trajectories are presented by Artemenko et al. [31]. The authors observed that a UAV spends a lot of time and energy making turns, once the vehicle has to decelerate, rotate and accelerate every time it performs these maneuvers. Thus, using the concept of Bézier curves, the algorithms modify conventional trajectories such as SCAN (back-and-forth), HILBERT, and LMAT, shown in Figure 8, by smoothing maneuvers along a given path. A more effective turning maneuver can be performed smoothing the movement with minimum deceleration. The authors compare the modified trajectories with the conventional ones and conclude that the new trajectories are able to reduce the amount of energy and time spent, keeping the level of the localization accuracy (LoLA).

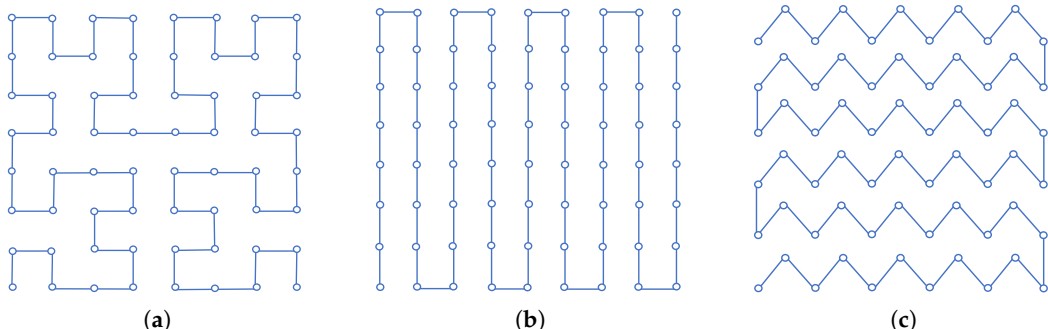

(**a**)                    (**b**)                    (**c**)

**Figure 8.** Conventional CPP paths: (**a**) Hilbert curves; (**b**) SCAN; (**c**) LMAT.

Forsmo et al. [41] employs the Mixed Integer Linear Programming (MILP) for coverage missions in rectangular areas involving UAVs. The waypoints distribution over a certain area considers the UAV on-board camera in order to obtain full coverage. The authors simplify the problem considering only rectangular obstacles and placing the aerial vehicles in distinct zones of the area, not using any type of area decomposition nor properly dealing with the collision avoidance issue among the vehicles.

Simulation experiments were performed to evaluate the proposed solution considering different cases with constraints, such as waypoint visitation order and camera range reduction.

Coverage missions may require a team of aerial vehicles working in a cooperative way in order to improve the task performance given the complexity and size of different scenarios. A cooperative coverage algorithm with critical time for rectangular areas using multiple fixed-wing heterogeneous UAVs is presented by Ahmadzadeh et al. [42]. The vehicles fly at distinct and fixed altitudes, such as 80 m, 90 m, 100 m, and 110 m, and with constant velocity. Furthermore, the vehicles present maneuverability restrictions and fixed cameras either in the front or in the left wing. An approach based on Integer Linear Programming is used to generate a solution considering these restrictions.

The paths of the UAVs with the frontal camera are basically circular, while the paths of the UAVs with the left side camera are composed of straight lines and left turns. When this vehicle turns right, the camera focuses on the horizon and does not capture any image of the coverage area or image resolution drastically decreases. The authors compare the proposed approach using the four fixed-wings vehicles against simple methods such as back-and-forth. Due to motion constraints and field of view, simple patterns such as back-and-forth presented a coverage of about 80% of the area of interest, while the proposed approach obtained 100% of coverage. The proposal was tested and evaluated in simulations performed in MATLAB and in real flights.

## 4. Exact Cellular Decomposition

The exact cellular decomposition can be adopted in the area of interest depending on the size and complexity of the workspace. Using this technique, irregular-shaped areas are split into sub-areas in order to reduce the concavities and simplify coverage. These sub-areas can be covered by single or multiple UAVs. In the former case, the CPP approach must concern the coverage path in each one of the sub-areas and the intermediate paths connecting those sub-areas. In the latter one, the CPP approach must worry about the relative capabilities of each vehicle in order to compute the size of each sub-area. Furthermore, a safety margin should be considered to prevent collision among the vehicles. First, we revise some CPP approaches for single UAVs using back-and-forth and spiral patterns for convex and concave areas. Next, we explore cooperative strategies dealing with multiple UAVs.

### 4.1. Single Strategies

An exact cellular decomposition approach considering concave polygonal areas is explored by Jiao et al. [43], Li et al. [44]. Initially, the workspace is decompounded in non-concave sub-areas through a minimum width sum approach exploring a greedy recursive method previously proposed by Levcopoulos and Krznaric [45]. Then, back-and-forth motions perpendicular to the sweeping direction, which is the minimum distance between an edge and a vertex, are performed in order to minimize the turning maneuvers [46]. Two sub-regions entirely adjacent and with the same sweep direction, obtained from the convex decomposition (from Figure 9a), are combined into the sub-region P4 to avoid unnecessary back-and-forth moves, as revealed by Figure 9b. It is also possible to change the motion direction from one sub-region to another in order to obtain a coverage improvement, as shown in Figure 9c. Finally, the optimal sequence of sub-regions is defined to join the final trajectory, as illustrated in Figure 9d.

Another coverage approach exploring the exact cellular decomposition for convex and concave areas is presented by Torres et al. [35]. The authors aim to capture pictures using aerial vehicles in order to achieve a 3D reconstruction. Convex polygons can be swept by BF motions according to the optimal direction. However, in more complex areas as concave polygons, it is needed to check if the mission can be performed in the same way with no gaps during the stripes, i.e., none of the stripes cross a part outside of the polygon. This particular case is illustrated in Figure 10a. When the path is interrupted, as shown in Figure 10b, an exact decomposition of the polygon is used to simplify the area creating sub-regions.

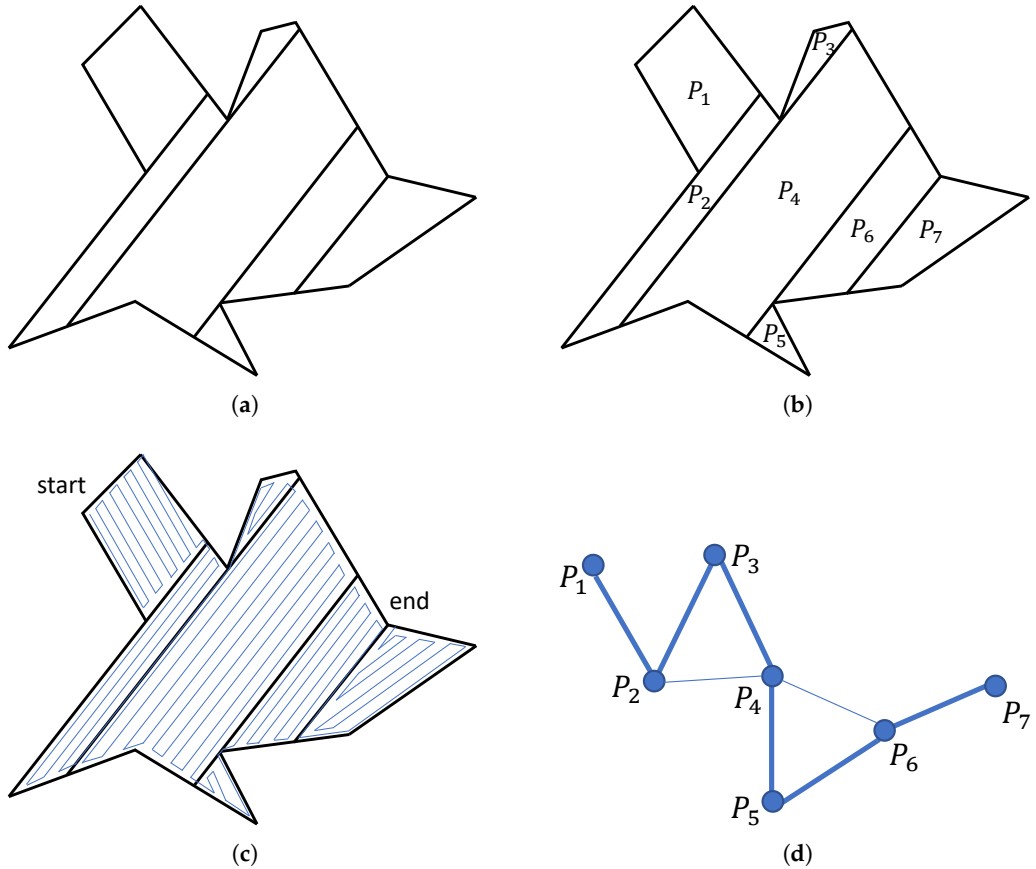

**Figure 9.** Decomposition and coverage of concave polygons: (**a**) Convex decomposition; (**b**) Sub-region combination; (**c**) Coverage path; (**d**) Undirected graph.

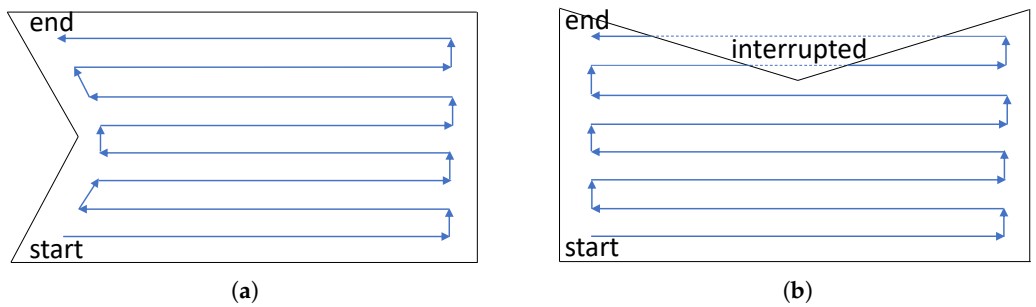

**Figure 10.** Coverage using back-and-forth pattern in concave polygons: (**a**) Non-interrupted path; (**b**) Interrupted path.

Once the optimal sweeping direction is defined for each sub-region, four different back-and-forth alternatives are explored considering two criteria related to the direction and the orientation. The former one explores if the coverage is going to follow the optimal motion direction or opposite way. The latter one considers the orientation of the first turning maneuvers, clockwise (left turn) or counterclockwise (right turn). The alternatives influence the transition distances, i.e., the distance between the last point of a given sub-region A and the first point of a given sub-region B. Permuting the sub-regions coverage order with the alternatives of each one, it is possible to minimize the transition distances and, consequently, minimize the path length. When the coverage is complete, the approach directly connects the final point with the first one using a straight line.

Exploring all permutations may consume an elevated computational time depending on the number of sub-regions. Thus, the authors use only the adjacent sub-regions for the transitions,

reducing the number of permutations. The proposed approach was evaluated in two scenarios. In the first scenario, a concave area is decomposed in five sub-regions pondering only four adjacencies. The authors drastically diminish the amount of permutations and the computational time spent generating a solution with an insignificant increment in the total distance. In the second scenario, the approach is compared to the one proposed by Li et al. [44] using the same area, as shown in Figure 11. The proposed approach decomposes the area into only four sub-regions, computing only 80 turning maneuvers against 87 of the original work.

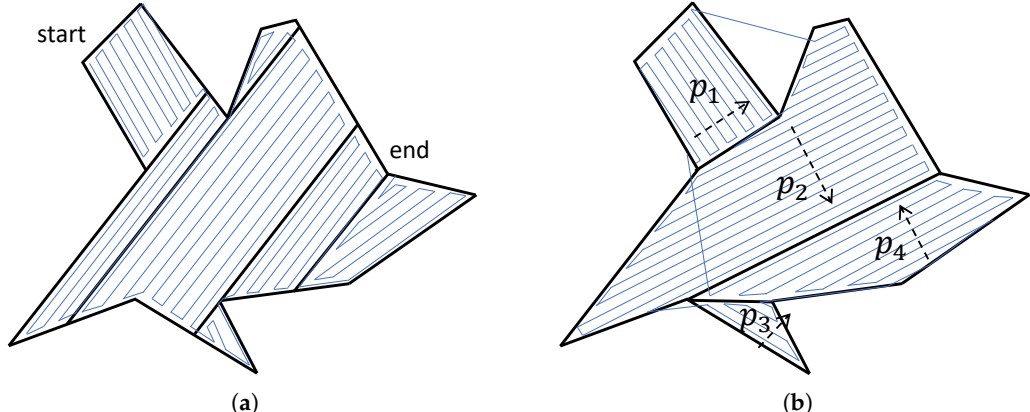

**Figure 11.** Comparison between the decomposition approaches in concave areas: (**a**) Convex decomposition [44]; (**b**) Concave and convex decomposition [35].

A coverage path planning technique for fixed-wing UAVs exploring wind to decrease the flight time is presented by Coombes et al. [47]. The authors incorporated the wind in the model to compute the coverage paths in a previous work [39] and extended their work by proposing a decomposition method to split the complex region into convex polygons. The area of interest is decomposed through trapezoidal decomposition exploring several rotations of the polygon. Cell recombination is employed using Dynamic Programming to merge cells into convex polygons. It is important to notice that this decomposition method also considers optional cells external to the region in order to find different decomposition with lower flight times.

The UAVs explore the area using back-and-forth motions perpendicular to the wind direction and are allowed to fly outside the area of interest. The initial and final waypoint of each straight line intersects the contour of the area and the UAV performs 180 degrees turning maneuvers to move from the final point of one line to the first point of the next one. This maneuver in the presence of wind is called a trochoidal turn. It consists of the shortest curve connecting the waypoints considering the fixed-wing restricted turning rate. The transition distance between adjacent cells is also considered during the path computation.

The authors present a cost function called Flight Time in Wind (FTIW) to compute the flight time needed by the aircraft to cover an area. The total time is the sum of the time to fly the straight lines, to perform the trochoidal turns, and to make the transitions between the cells. The proposed method (FTIW) is compared with previous techniques that aim to minimize the number of turns (NT) [44] and the sum of the altitudes (MSA) [46]. Using a real field and several random polygons generated by a Monte Carlo simulation, the authors claim that their FTIW approach overcomes both previous approaches considering the flight time needed to execute coverage.

Xu et al. [30] and Xu et al. [48] present an optimal coverage algorithm for fixed-wing UAVs able to avoid flights in obstacle-regions with arbitrary shapes and previously explored regions. The area of interest is decomposed into a simple set of cells using Boustrophedon Cellular Decomposition (BCD), originally proposed by Choset and Pignon [49], in an offline phase through a bitmap representation. The BCD is an exact cellular decomposition able to work with non-polygonal obstacles, which presents more efficient coverage paths than the trapezoidal decomposition. From the decomposition, an

adjacency graph can be built with the vertices representing the cells and the edges connecting the adjacent cells, as illustrated in Figure 12.

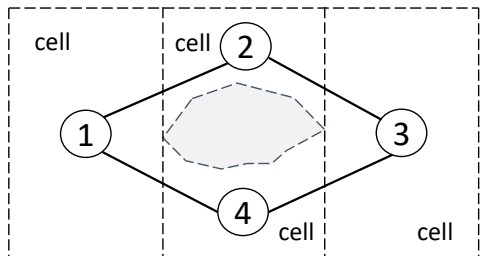

**Figure 12.** Area of interest with obstacles decomposed into cells and the graph representation.

Cells are explored using back-and-forth motions in an online phase. The order between the cells follows a Eulerian circuit with start and end at the same vertex. Depending on the next cell location, it is necessary to go through sub-regions previously explored in the scenario. Thus, a more efficient technique is proposed to eliminate the disjunctions adding an extra sweep line. This line guarantees the continuity of the path when it reaches the border of the cells and prevents repeated explorations. The original and the modified approach were applied in simulations and real flights. Both were evaluated regarding the total path length and the time to perform the coverage. The method proposed by the authors was 10% more efficient in both criteria.

A convex decomposition method to split complex shapes into smaller shapes and transform concave shapes with sharp edges into convex shapes is presented by Öst [29]. The author explores the back-and-forth and the spiral motion, as illustrated in Figure 13, combining them with the proposed area decomposition technique. The authors claim that back-and-forth pattern with no area decomposition presents reliable outcomes with respect to different combinations. This confidence is because all maneuvers have 90° turns, which allow us to predict the pattern after four moves. Despite generating slightly longer paths, this pattern is capable of handling complex shapes without losing coverage. However, the algorithm consumes a considerable time testing all different rotations in the polygon between 0° and 180° to find the optimal motion direction.

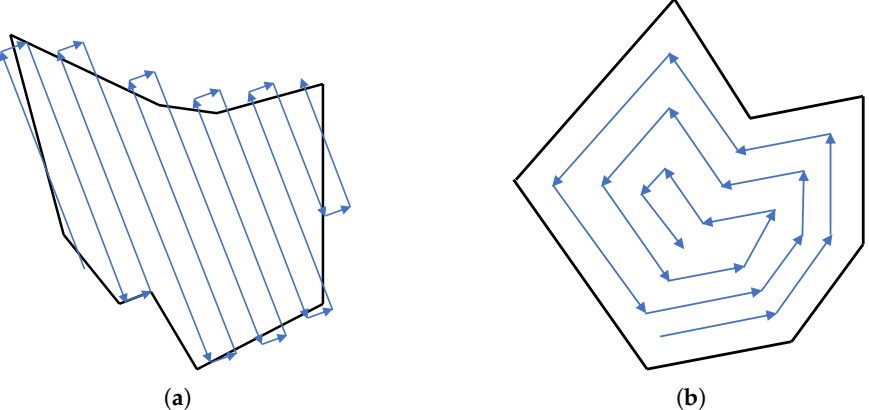

(**a**)        (**b**)

**Figure 13.** Simple flight patterns in polygonal areas: (**a**) Back-and-Forth; (**b**) Spiral.

Shortest paths are generated with spiral pattern in rounded shapes with large inner angles, according to Öst [29]. However, sometimes the pattern does not conclude the coverage in complex areas. Mixed variations comprising patterns and decomposition are able to generate a path with minor distance, but these combinations do not always deal with all the instances in a good way. It has proved to be really effective only when the area has too many protrusions in different directions, such as a star shape. In some cases, the decomposition may contain more vertices with a large number of small inner angles, generating self-intersections during the junction of the areas.

*4.2. Cooperative Strategies*

Cooperative strategies employ multiple UAVs to cover an area of interest. This type of strategy is usually applied when the workspace is too large to be covered by a single UAV. Depending on the complexity of the problem, the coverage approach may only split the area into sub-areas and plan coverage paths individually for each UAV. More complex approaches deal with motion synchronization, decentralized information and communication issues, and different levels of local priority.

### 4.2.1. Back-and-Forth

A cooperative strategy in a convex polygonal area using a team of heterogeneous UAVs is presented by Maza and Ollero [34]. A ground control station decomposes the area and assigns the resulting subregions to each vehicle considering the relative capabilities and initial locations. Each vehicle computes the back-and-forth motions aiming at minimizing the number of turning maneuvers. Figure 14 presents the area of interest decomposed with three vehicles and the respective coverage paths. The distance between the parallel stripes is related to the camera footprint of each vehicle. A reconfiguration process is employed when a vehicle fails, so the area is partitioned again and the remaining vehicles should recompute their sweep directions.

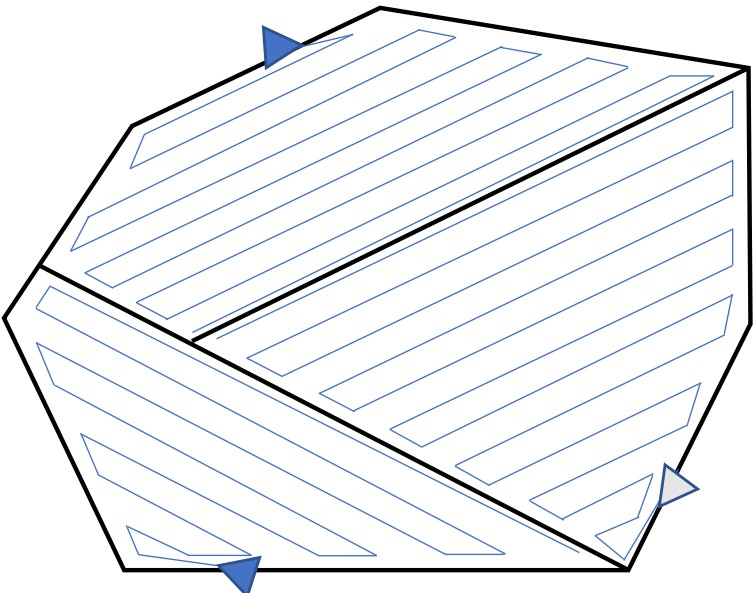

**Figure 14.** Cooperative coverage in convex polygon area using a team of heterogeneous UAVs.

### 4.2.2. Spiral

A spiral CPP algorithm for missions in coastal regions using multiple heterogeneous UAVs is explored by Balampanis et al. in several works. The authors discretize the workspace considering the sensing capabilities of the aerial vehicles using a Constrained Delaunay Triangulation (CDT) [50] introduced by Balampanis et al. [51] and Balampanis et al. [52]. The authors state that the classical grid decomposition creates regular square cells that are partially over no-fly zones or outside the workspace. Thus, the CDT provides triangle cells within the area of interest matching almost the exact shape of the area.

In order to generate more uniform triangles, they applied the Lloyd optimization [53]. This technique approximates the cell angle to 60 degrees, enhancing the uniformity. Then, a spiral algorithm previously proposed by Balampanis et al. [54], and improved by Balampanis et al. [55] by introducing a smoothing parameter, is used to generate the coverage paths in the resulting sub-areas. The proposed strategy has been tested in Software-In-The-Loop simulation. A further analysis regarding coverage patterns is performed by comparing the spiral-like pattern using the CDT/Lloyd optimization (triangle cells) with

the classical grid decomposition (square cells) using back-and-forth motions. The authors claim that their approach is capable of covering fully a given area with a smoother trajectory without entering into NFZ or going outside the area. However, the approach presents longer coverage paths with a higher number of turns compared with the back-and-forth.

### 4.2.3. Line Formation

A cooperative coverage strategy exploring line formation is proposed by Vincent and Rubin [56] to detect intelligent targets moving in dangerous environments. The authors consider that the targets try to deliberately escape from the search performed by the vehicles inside a rectangular area. The mission is based on five criteria, such as maximize the probability of detecting a target, minimize the tracking time and the number of aerial vehicles employed in the mission, provide robustness on the assumption of a failure of one or more UAVs and minimize the amount of information shared among the vehicles.

The vehicles are organized in a line formation and execute long straight movements, as illustrated in Figure 15. As vehicles explore the area in a close formation, the communication among them is simplified and it favors the continuity of the mission in case of a failure in one of its components. There are two types of control messages exchanged between the vehicles, maintenance messages and update messages. The maintenance message is periodically exchanged between the vehicles and if one of them fails to transmit the message, the adjacent vehicles assume that the vehicle is out of action. The update messages are exchanged by the vehicles to achieve an agreement for the pattern reconfiguration. The success of the coverage pattern depends on the return to the previously explored neighborhood areas, once the target may move to such areas in an attempting to escape from the sensors of the vehicles.

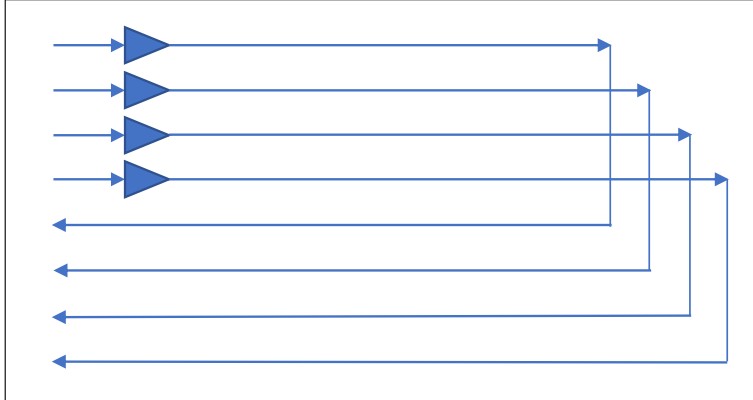

**Figure 15.** Cooperative coverage strategy in rectangular areas of interest with intelligent targets.

### 4.2.4. Decentralized Technique

A decentralized algorithm for partitioning rectangular areas during surveillance missions is presented by Acevedo et al. [57]. Homogeneous vehicles distribute the sub-regions using a one-to-one coordination technique and explore adjacent areas. The vehicles have a short range of communication, as shown in Figure 16, and must share information passing by near points in a synchronized way. The sub-regions are equally distributed with paths of the same size generated by the sub-perimeter method. This method developed by the authors uses the information about the area to generate an interior similar region, such that the maximum distance from the inner zone to the border of the area is less or equal than the coverage range. The system is adaptable to modifications in the team, such as a departure of a vehicle for repairs or a battery recharge. The main goal is to develop a cooperative patrolling strategy to optimize the observation interval among consecutive attempts in any spot and minimize sharing time of detected information with the other members (latency).

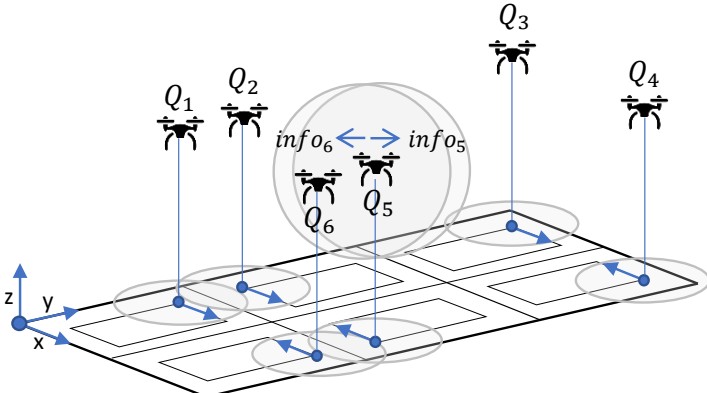

**Figure 16.** Decentralized algorithm for surveillance task in rectangular areas with information exchange using a team of homogeneous UAVs.

More recently, Acevedo et al. [58] extend their approach for surveillance in irregular-shaped areas with heterogeneous vehicles. The irregular area contains obstacles and irregular borders, thus it is discretized into regular regions, as shown in Figure 17a. The numbered black rectangles in this figure represent forbidden flight zones used to delimit boundaries and indicate obstacles. In this context, instead of partitioning the area of interest, a single path that covers the entire region is segmented and distributed to the vehicles, sharing information among the neighbors. Faster vehicles cover larger path segments and all vehicles invert the patrolling direction at the end of each part. In simulations, an urban scenario with constructions is considered, as illustrated in Figure 17b, with four vehicles flying in low altitude, while avoiding obstacles. The system is able to adapt itself to the entrance and the exit of vehicles during the mission.

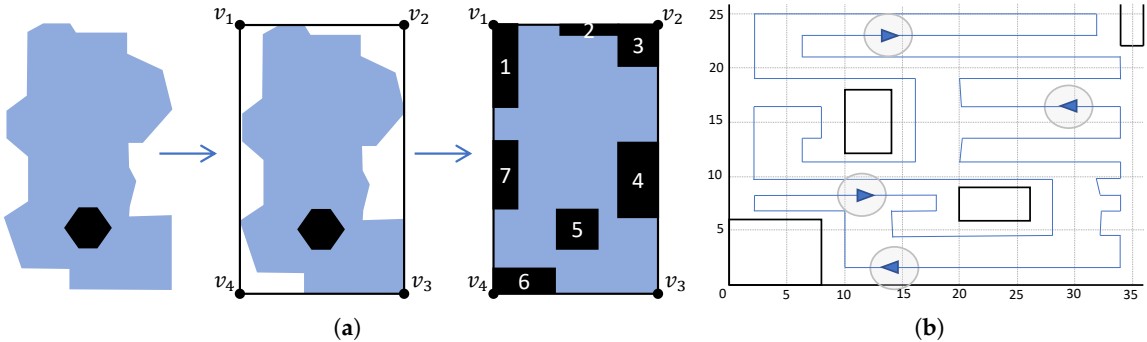

(**a**)　　　　　　　　　　　　　　　　　　　　　　　　　　　　　　(**b**)

**Figure 17.** Surveillance mission in irregular-shaped areas with path segmentation using a team of heterogeneous UAVs: (**a**) Irregular area approximation; (**b**) Segmented single path.

Finally, Acevedo et al. [59] explore the one-to-one coordination approach using a grid-shape area partition strategy. The area of interest is partitioned into non-overlapping sub-regions monitored by heterogeneous vehicles following distinct paths. This technique allows the vehicles to self-adjust the partitioning according to their maximum capacities, maintaining the synchronization in a distributed and decentralized way. Furthermore, this solution is able to rearrange primary conditions, such as area shape and vehicles capacity.

### 4.2.5. Local Priority

A continuous coverage and decomposition approach for convex polygonal areas with local priority is proposed by Araujo et al. [60]. The workspace is decoupled in minor areas with a sweeping method, where a portion of the area should be assigned to each vehicle according to its relative capability, such as the amount of area covered per unit of time. The authors developed a method to generate an optimal number of stripes inside each sub-region, considering the kinematic constraints of

the fixed-wing vehicles. Using a diameter function that describes the polygon altitude, a perpendicular optimal sweep direction can be obtained which minimizes the number of stripes and, consequently, the number of turning maneuvers. The stripes have the same width of the on-board camera footprint and an overlapping between the stripes is necessary due to positioning errors and slightly variations in the trajectory.

The authors discard the use of some flight standards, as simple back-and-forth and spiral motions, claiming that these patterns can not neither deal with different local priorities nor perform multiple visits in specific areas. Thus, the authors propose a back-and-forth/zamboni (Zamboni refers to the machines that repair the ice in hockey arenas.) flight pattern, illustrated in Figure 18. The proposed flight pattern allows visiting previously explored stripes since the degree of uncertainty of a locality raises as time passes since the last visit. They also consider that the localities present different degrees of priority managed by a human operator. So, after completing coverage in one of the stripes, the vehicle must select the next stripe considering the uncertainty and the priority.

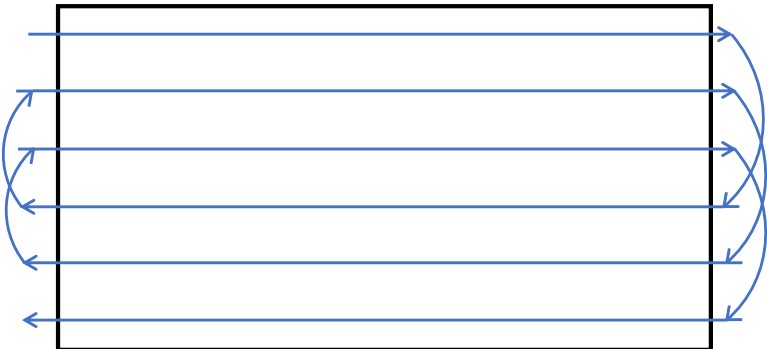

**Figure 18.** Back-and-Forth/Zamboni flight pattern with local priority and degrees of uncertainty for continuous coverage missions.

## 5. Approximate Cellular Decomposition

A coverage trajectory is usually planned before its execution, in an offline phase, considering that the aerial platform has full knowledge of the workspace to be covered. However, in some cases the aerial vehicle has to interleave between planning and execution, gathering information through its sensors to build an internal map as it moves around the scenario. In both cases, the approximate cellular decomposition can be employed to discretize the area of interest into a grid, while a grid-based solution can be used to perform the coverage mission. First, we revise some complete algorithms proposed to deal with irregular-shaped areas, considering approaches based on a single (as Valente et al. [32]) and based on multiple UAVs (as Barrientos et al. [10]) with full information about the scenario. Next, we explore some cooperative bio-inspired approaches in order to deal with environments containing only partial information.

### 5.1. Full Information

Back-and-forth flight pattern is usually employed in applications like agriculture, but this type of motion generates inefficient trajectories considering areas of interest with an irregular shape. A coverage path planning approach for image mosaicing in precision agriculture with irregular-shaped fields is proposed by Valente et al. [32]. In this approach, named gradient-based, the area of interest is discretized into a regular grid using the approximate cellular decomposition, as shown in Figure 19. Each cell represents a waypoint of the path and its size depends on the picture dimensions of the on-board camera. The grid configuration is attained through image demands and image sensor characteristics. The decomposed area is converted to a regular graph numerically labeled by the Wavefront algorithm, which is a flooding algorithm that marks the neighborhood adjacency of cells.

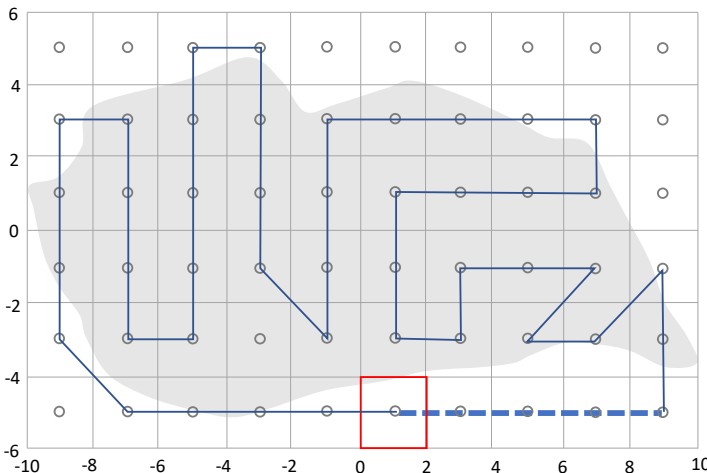

**Figure 19.** Grid-based method in an irregular-shaped area of interest.

A Deep-limited search (DLS) [61] is applied to discover a full path without revisiting previously explored nodes. With the DLS, the exploration length is restricted to the quantity of vertices and do not gets stuck in loops or revisits a node. A backtracking procedure is also employed to solve issues such as the choice among neighbors of equal values. The proposed method enables a simplistic and faster solution to achieve a near-optimal results in complex areas with certain constraints.

Another approach exploring the Wavefront algorithm and the approximate cellular decomposition for coverage in agricultural areas is presented by Nam et al. [62]. The coverage path is obtained over an area of interest labeled according to the Wavefront, as shown in Figure 20a, and smoothed through a cubic interpolation algorithm, as illustrated in Figure 20b. Different from Valente et al. [32], the authors present a novel optimization criterion for the mission execution time based on the path length and the number of turning maneuvers.

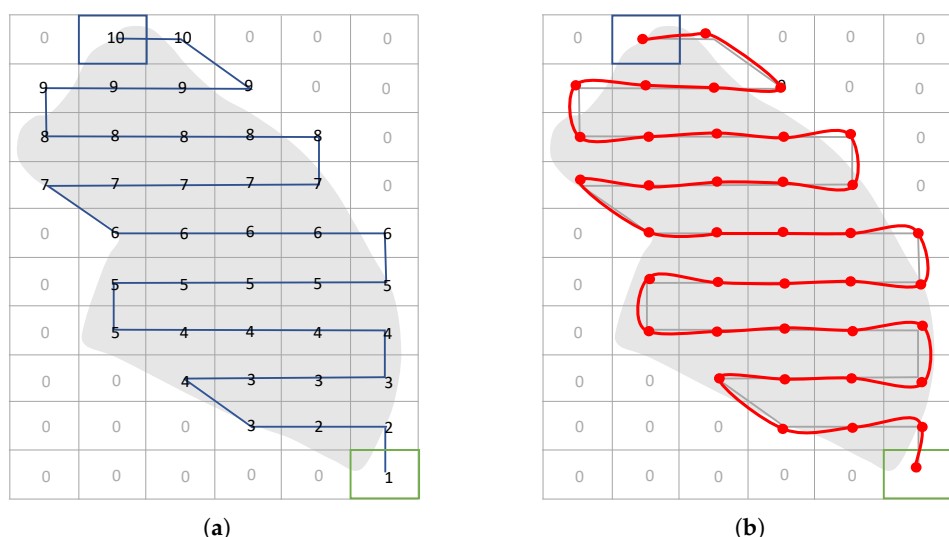

**Figure 20.** Optimal path and coverage trajectory: (**a**) Wavefront; (**b**) Cubic Interpolation.

An optimal CPP algorithm with a quadrotor UAV is presented by Bouzid et al. [63]. The area of interest is represented as a map containing Points Of Interests (POIs). The UAV should perform a minimum path connecting the POIs and avoiding obstacles with distinct formats to guarantee complete coverage in the area. The mission is planned in two steps. The algorithm computes the travelling cost to explore the neighborhood and then determines the sequence of points that should be visited in order to minimize the total distance. After visiting the POIs a single time, the vehicle should return to

the initial position. In this way, the problem is treated as the Traveling Salesman Problem (TSP) and the overall shortest path can be computed using a Genetic Algorithm (GA). The authors consider the accumulative Euclidean distance among the points as the performance metric. Besides, they consider the energy consumption to be constant during the whole mission, measuring it in terms of time.

In real-world applications, the quadrotor may need to charge or replace the battery a couple of times while performing a task considering its limited on-board energy. Thus, the authors explore a different possibility inspired on the Vehicle Routing Problem (VRP). This solution finds the minimum group of shortest trajectories in cases with only one or several initial positions. In this scenario, the aerial vehicle performs the designated task and keeps coming back to the base station every time it needs to recharge its battery. At the same time, the UAV also downloads the acquired information during each part of the mission.

A coverage approach for precision agriculture involving a team of heterogeneous quadcopters is discussed by Barrientos et al. [10], including two main phases: task subdivision and allocation, and coverage path planning. In the first phase, using a negotiation protocol, the subdivision of the area and the allocation of the resulting sub-regions are simultaneously accomplished in a distributed way. In this phase, the vehicles have to analyze the cost and the recompense of a task, i.e, cover a certain subregion.

Considering its internal parameters, the vehicle evaluates the cost of a task execution, the initial cost to move from the current location to the search area, certain restrictions as NFZ, turning angles and embedded sensors, and the prize related with the mission accomplishment. The objective function sums several terms with different weights, including the task dimension and the distance between the starting point to the mission location. Furthermore, a penalty can be applied if one task superposes another or the exploration goes beyond the general area.

In a second phase, the area is discretized into a regular grid using approximate cellular decomposition. Each vehicle uses a Wavefront algorithm to create a coverage path for its sub-region minimizing the flight time, the quantity of turning maneuvers and the amount of revisited cells. Also, the vehicles keep a constant altitude to guarantee a desired resolution according to the field of view of the on-board camera. Finally, the authors present a control system to improve the vehicle's altitude stabilization during high-speed maneuvers. Experiments were performed with three vehicles in a vineyard field, characterized by its irregular-shaped format and its changeable altitude profile. The sub-regions boundaries delimit a safe zone, in which the vehicles should not get in to avoid collisions, as illustrated in Figure 21.

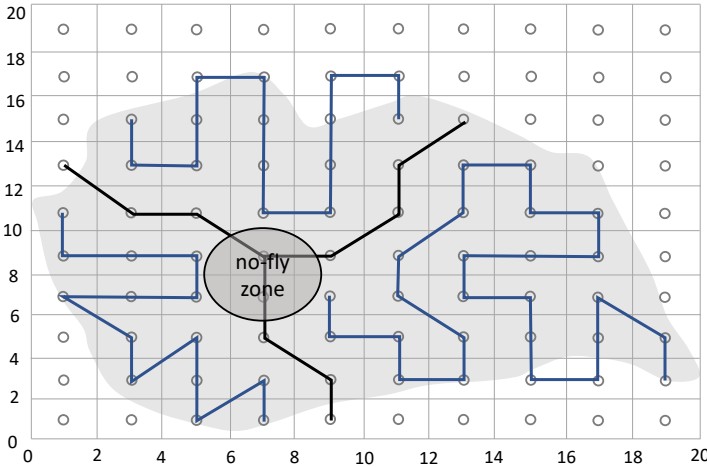

**Figure 21.** Coverage path planning in irregular-shaped areas containing no-fly zones in the subregions bounds for collision avoidance.

Focusing also on precision agriculture, a meta-heuristic algorithm named Harmony Search (HS) is proposed by Valente et al. [64] to minimize the number of turning maneuvers in irregular areas. The proposed algorithm is based on jazz musician's improvisation and its main body is a Harmony

Memory (HM). The HM consists of a matrix, where the lines are composed by vectors comprising possible solutions, while the columns represent decision variables. The cost-function result is placed in the last column.

Following the approach proposed by Valente et al. [64], the matrix initialization is performed randomly and an iteration known as improvisation begins when the first generation is complete. New vectors are created through exchanges between neighboring cells according to a certain probability. If a new harmony has an improvement in comparison to the worst solution, it replaces the old vector in the matrix. Otherwise, the matrix would remain unchanged. The authors compared the HS approach against the Wavefront algorithm employed by Barrientos et al. [10] using the same scenario with three subregions, as shown in Figure 22. The HS achieves a smaller number of turns than the Wavefront, but a longer computational time. However, the authors do not consider computational time as a problem, once the planning is performed offline.

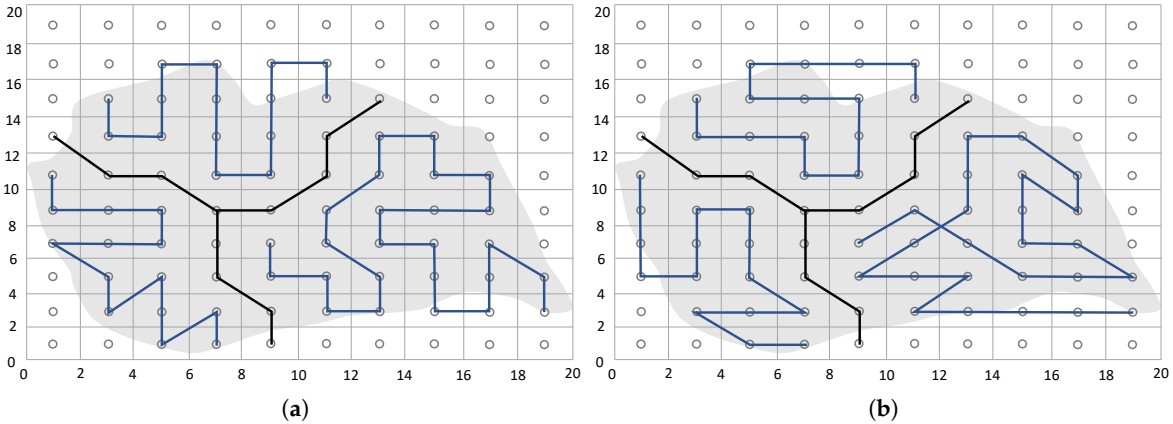

(a)  (b)

**Figure 22.** Comparison between different coverage path planning approaches in irregular-shaped areas using three UAVs: (**a**) Wavefront Algorithm [10]; (**b**) Harmony Search [64].

More recently, Sadat et al. [65] and Sadat et al. [66] propose approaches for online non-uniform coverage path planning. In this context, the vehicle can change its altitude during the coverage according to the importance of each part of the area. The authors use a tree structure to deal with grids considering different resolutions. The closer the node is to the leaves, the higher is the resolution. Sadat et al. [65] introduced three methods to explore the tree: Breadth-First strategy, Depth-First, and Shortcut Heuristic, while Sadat et al. [66] proposed an Hilbert-based approach, as shown in Figure 23, comparing it with the previous strategies and the lawnmower pattern. When visiting a zone of interest, the resolution is increased by traversing down the tree. In this way, the area that was being explored by a parent node, now is being fully covered by their child nodes at an increased resolution. On the other hand, if the zone is not interesting, the search goes up in the coverage tree and decreases the resolution. Therefore, a single vehicle moves around the area exploring zones at different altitudes to perform the mission.

High-resolution aerial sensing with multiple heterogeneous UAVs for non-convex areas is discussed by Santamaria et al. [67]. The vehicles present different coverage range and image sensors, so the area of interest is discretized into a grid with cells of different sizes through the approximate cellular decomposition, as illustrated in Figure 24. Initially, a roughly estimation determines the portion of the area to be explored by the UAVs considering their on-board sensor footprint. A flooding technique selects a starting position and extends the neighborhood area until it reaches the estimated amount of cells. Places located out of the area or within NFZ are invalid cells. After the first completed round, the unsigned cells are addressed to the nearest area and a re-balance is executed to uniformly distribute the cells.

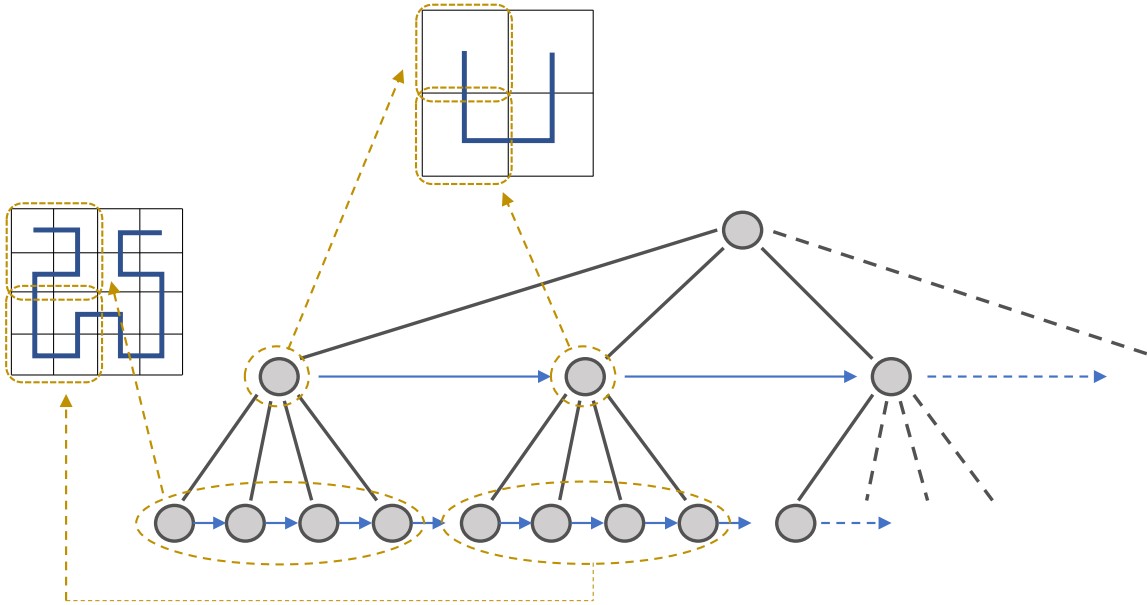

**Figure 23.** Coverage performed at distinct altitudes according to the importance of each zone. The closer the node is to the leaves, the higher is the resolution.

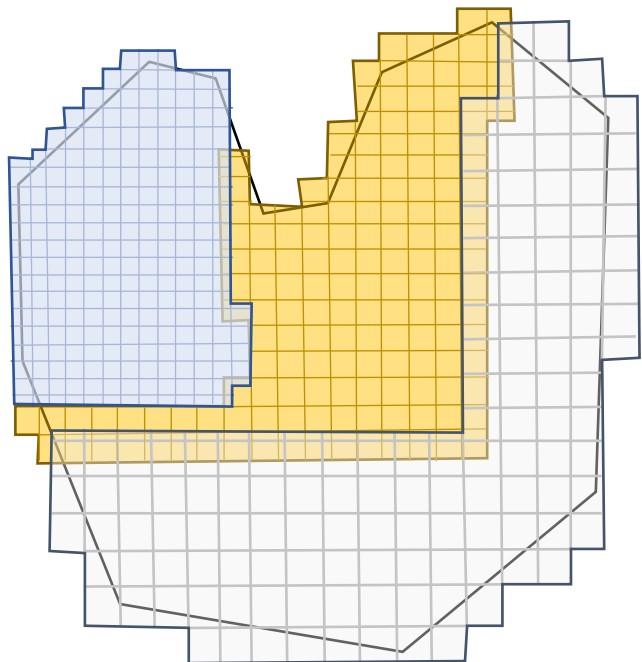

**Figure 24.** Area of interest discretized into a grid through the approximate cellular decomposition with cells of different sizes.

Following the approach presented by Santamaria et al. [67], flight paths to cover each of the sub-areas are computed using an algorithm to find long-distance segments of free cells, called strides. The algorithm computes the strides for all unexplored adjacent cells of the actual position, selecting the longest line. The last cell of the selected line turns into the current cell, repeating the process. When there are no unvisited neighbor cells, the algorithm generates trajectories from the actual location to the closest unexplored cells, selects the minor path and adds the corresponding cells to the coverage plan. The shortest distance is also computed to return to the landing location. The proposed approach was integrated with the AMFIS [68], which consists of ground control station for real-time vehicle controlling and monitoring.

## 5.2. Partial Information

Biological systems are able to adapt, evolve and survive in natural environments characterized by quick modifications, elevated uncertainty, and restricted information [69]. Since such natural systems are robust and sophisticated, they have been replicated as computational systems over the last decades in order to solve complex optimization problems and application topics [70].

Thus biologically-inspired approaches consisting of algorithms based on fundamental aspects of natural intelligence have emerged, such as behavioral autonomy and social interaction, evolution and learning [70]. Considering the CPP problem with aerial vehicles, several authors have explored different approaches in the literature, including real-time search methods [36], random walk [71], cellular systems [72–74], evolutionary computation [75,76], and swarm intelligence [77–79]. Coverage with uncertainty considering information points is also addressed [80–85]. Most of the approaches are pheromone-based and explore the natural behavior of ants to guide the vehicles through a grid-discretized scenario.

### 5.2.1. Pheromone-Based Methods

Pheromone-based methods alternate between planning and execution, allowing a fine adjustment when performing such operations. They are inspired by the actions of real ants that employ chemical tracks to orientate navigation. These methods represent the workspace as a grid and use an *u-value* associated to each environment cell. This value represents the pheromone marks left by the vehicles moving through the scenario, i.e., the quantity of visits on each location [86]. Pheromone can be inserted in and/or taken off from a local, evaporate through time and/or be propagated to the neighborhood, according to the adopted strategy. Different flavors of pheromone can represent different kinds of information, while some types of pheromone can attract or repel vehicles.

The field of view of the vehicles is usually restricted to the adjacent cells, enabling sensing and motion actions only in the cells immediately orthogonal, as shown in Figure 25a. The scenario may be discretized as a connected graph during the search, as illustrated in Figure 25b. The versatility of some of these approaches to deal with different applications in a military context is discussed by Sauter et al. [87]. These methods have been proposed to solve CPP problem with land vehicles. As these methods present a low computational cost, any autonomous vehicle may employ them appropriately [86]. However, most of the previous studies are limited only to simulations [72,86,88] and just a few methods have addressed contexts with aerial vehicles in real-world scenarios [36,74].

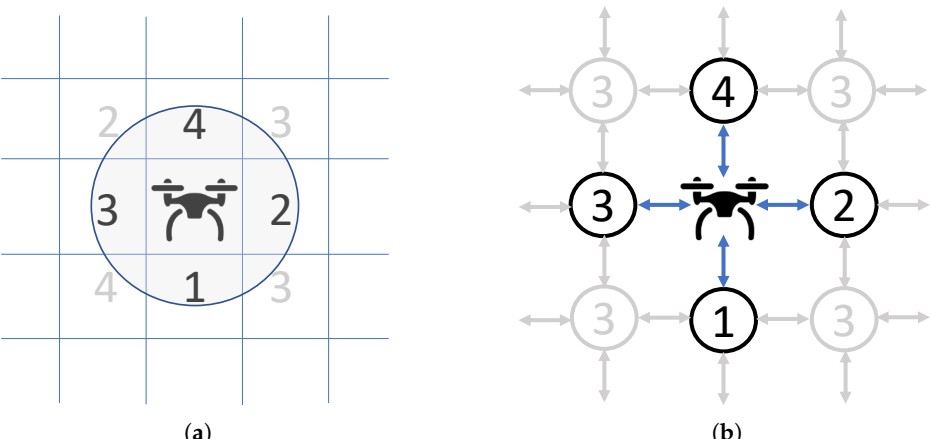

|     |     |
| --- | --- |
| (**a**) | (**b**) |

**Figure 25.** Environment representation in pheromone-based methods: (**a**) Field of view; (**b**) Connected graph.

### 5.2.2. Real-Time Search Methods

A performance of real-time multi-robot coverage algorithms is analyzed by Nattero et al. [36]. The authors explore classical heuristics, such as Edge Counting [89], PatrolPGRAPH* [90], Node

Counting [91], and Leaning Real-Time A* [92]. The algorithms were evaluated in simulations with land robots in terms of the required coverage time and, implicitly, the energy required to perform a mission. Four types of experiments were executed varying aspects, such as grid size, grid topology, quantity of robots and required visits.

The Node Counting overcomes the other solutions during the simulations and it was implemented in a real system with a quadcopter and a hexacopter. An off-board control implemented using Robot Operating System (ROS) [93] is applied to guide vehicles sending target positions and periodically receiving the localization information of the vehicles. Control dynamics and vehicles localization are executed on-board through ETHNOS framework [94]. The decision process is centralized in the off-board station, which communicates with the vehicles using an ROS/ETHNOS interface. To apply virtual pheromones in an aerial scope, a centralized process is required, which makes the system less robust to failures once all vehicles depend on the communication with the ground control station to perform their tasks.

### 5.2.3. Random Walk

An approach for field coverage and weed mapping is presented by Albani et al. [71]. The proposed approach consists of an exploration strategy that uses a reinforced random walk to detect the existence and the amount of weeds using UAV swarms. Each individual UAV splits the search plane into two parts, preferring to explore the semi-plane ahead according to a utility value. This value is defined by the vehicle's momentum and the angular difference between the current cell and the scanning direction, influencing the decision about the next cell to visit. The value is elevated for locations line up with the momentum vector. Additionally, the influence from neighboring UAVs reflects on the performed motions, randomly guiding the vehicles to poorly explored zones. This is carried out by computing a repulsion vector. The vehicles also exchange information among them in order to prevent previously surveyed covering zones. Finally, the swarm convokes their members in direction to promising zones to execute the weed mapping using an attraction vector.

### 5.2.4. Cellular Automata

An algorithm based on cellular automata, originally applied by Zelenka and Kasanicky [72] to coordinate robots in land coverage tasks, is employed by Zelenka and Kasanicky [74] to control two quadcopters in exploration and monitoring tasks. The adaptive decentralized system works with a shared memory represented by the environment. Virtual marks simulate the pheromones to coordinate the vehicles. A ground control station (GCS) splits the environment into virtual cells, monitors the position of the vehicles and prevents collisions sending new coordinates. Despite being considered as an adaptive decentralized system by the authors, the vehicles share a global memory using the GCS and do not make the decisions alone.

The authors also explore the strategy considering the degradation of the environment's virtual marks to emulate the loss of communication [73]. The process of evaporation or degradation consists of reducing the amount of pheromones of a certain place when it is not visited for a while. However the degradation drops the algorithm efficiency and causes a lot of collisions between vehicles. The accumulation of a high amount of pheromones in small places may also cause coverage problems. When the aerial vehicles pass over the same locations, several times in short periods of time, they prevent the evaporation process and block other vehicles to perform the mission.

### 5.2.5. Coverage with Uncertainty

A waypoint planning algorithm for surveillance is presented by Lim and Bang [81]. The area of interest has the form of a hexagonal grid and contains Information Points (IP), as shown in Figure 26a. Each IP works with a certainty value that quantifies the information confidence. When the certainty value is significant, the IP contains trustworthy information and the exploration in the surrounding

areas is not necessary. However, this certainty drops along the time with the absence of observations. Lower values denote poor information and demand a new observation in the location.

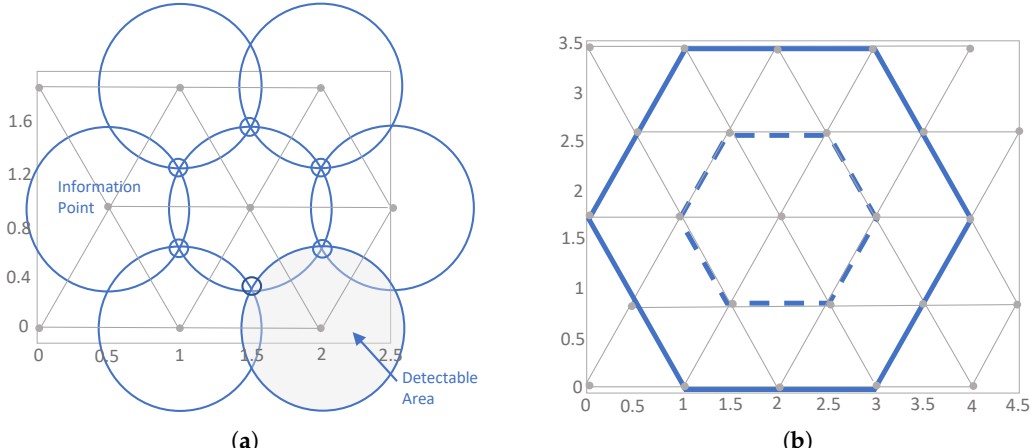

**Figure 26.** Hexagonal grid method with information points representing the uncertainty: (**a**) Information Points; (**b**) Certainty calculation with IPs.

The algorithm uses cost functions such as certainty, distance and interest level to guide the vehicles and select the next points to be explored. In a previous work [95], the authors present details about the cost functions. Regarding the certainty cost function, it is necessary to consider the average certainty of neighboring IPs, guiding the vehicle not only to the point but to the region with the lowest certainty. The A* algorithm is employed to generate the path to reach the next point chosen using cost functions. As a cooperative surveillance mission, a sub-region is assigned to each vehicle to avoid coverage overlapping.

Khan et al. [80] presented an approach for merging distributed information in cooperative search. The multiple UAVs have several restrictions regarding the sensing capabilities and information trading. Besides, false alarms with a certain probability are also considered by the sensors. The mission consists of a group of small-scale UAVs searching for target locations within a rectangular area discretized into square cells. The vehicles can perform motions in four different directions or remain in the present location. The individual vehicles can sense the area and update the search map, while sharing local information with each other. Fusing their local maps, the aerial vehicles are capable of speeding up the search, while improving the sensing capabilities in relation to approaches without a cooperative strategy.

Popović et al. [82] introduced an Informative Path Planning (IPP) for weed detection in precision agriculture using UAVs. The authors use a fixed-horizon approach for adaptive planning, alternating between plan execution and replanning. They employ a two-stage replanning, where the 3D path of an aerial vehicle is constantly refined, while respecting motion restrictions. This optimization is obtained using the Covariance Matrix Adaptation Evolution Strategy (CMA-ES).

The authors present a multi-resolution IPP for terrain monitoring with aerial platforms [83]. The proposed approach is built upon the method established in the previous work [82]. But instead of a binary classification of weed occupancy, the method focuses on biomass monitoring. The method builds a probabilistic map with all the visual data gathered in flights at distinct altitudes. The method proposed by Popović et al. [82] and further explored by Popovic et al. [83] is evaluated in simulation with respect to the back-and-forth and the state-of-the-art IPP algorithm. The authors also run their IPP approach in an AscTec Pelican UAV, emulating the weed classifier through the use of AR tags [82] and on a DJI Matrice 100, mimicking the vegetation monitoring on painted green sheets [83].

A learning-based algorithm for persistent surveillance problem using UAVs is addressed by Ramasamy and Ghose [84]. The problem can be described as a continuous CPP, where the UAV should explore all the locations of an certain area from time to time, while minimizing the interval between those visits. The problem may be more complex considering the necessity of increasing or reducing the

frequency of visits depending on the zone profile—an interesting one or a risky one. Ramasamy and Ghose [85] extend their work dedicating more attention to the preferential surveillance in a known area. They explore an approach considering different ways of quantitative priority specifications. Besides, they present further simulation results to support a more detailed analysis.

5.2.6. Genetic Algorithm

A combination of digital pheromones and evolutionary strategies for coordination of multiple UAVs is explored by Paradzik et al. [75]. The workspace is decoupled in rectangular sub-regions assigned to each vehicle using a Genetic Algorithm (GA), originally proposed by Holland [96]. The population individuals carry information regarding the vertices' coordinates and the width/length of the rectangular sub-regions. The individuals go through the selection, the reproduction and the mutation phases for generating new individuals during the natural evolution process of the algorithm. A fitness function evaluates the best individuals (sub-regions) based on digital pheromones containing information about the region. Figure 27 shows an example of sub-region created using the GA.

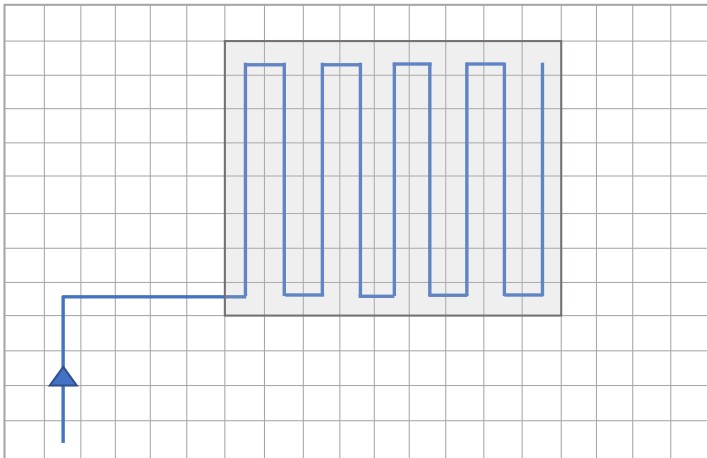

**Figure 27.** Area of interest decomposed into rectangular subregions using GA and covered using the back-and-forth flight pattern.

The authors use two pheromone flavors: search and path. Search pheromone corresponds to the uncertainty and the need for coverage of a region, while path pheromone indicates the locations that are already included in the path of some vehicle. Path pheromone is applied to prevent collisions and decrease the revisiting probability. The aerial vehicles should pursuit trajectories aiming at maximizing the amount of search pheromone, while minimizing the quantity of path pheromone. The pheromone-based GA only divides the environment and the vehicles employ back-and-forth motions to cover each sub-region. In addition, each vehicle uses the A* algorithm to plan a path from its initial location to the nearest vertex of the assigned sub-region.

Considering another effort based on a GA [76], a coverage path planning approach with 3D structure mapping is proposed to handle scenarios with obstacles and buildings. The area of interest is a polygon discretized as a grid containing buildings (yellow), tall vegetation (green) and forbidden flight zones (red), as shown in Figure 28a. The cells are labeled with the following values: 0 for free areas, $-1$ for forbidden zones or outside zones, $-2$ for tall vegetation that demands altitude adjustments and $-3$ for buildings. Using a GA, the coverage path is generated considering only the free spaces and the areas with vegetation below the altitude flight.

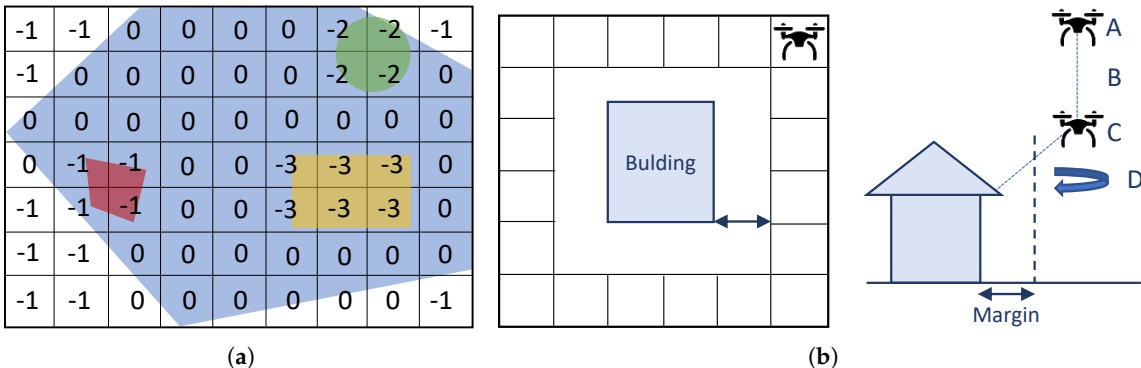

**Figure 28.** Grid representation of the area of interest and main steps of 3D mapping method: (**a**) Labeled grid area; (**b**) Structure mapping.

During the path execution, adjacent cells containing buildings may be detected, triggering the 3D mapping. In this case, the vehicle stops the coverage and surrounds the building at a certain distance to photograph it, following the next steps: hovers at the current altitude (A), goes up/down according to the height of the building (B), changes the camera angle (C) and flies surrounding the building (D), as shown in Figure 28b. Once the mapping is completed, the vehicle continues the original coverage coverage path. Figure 29 illustrates a complete path avoiding forbidden flight zones, while surrounding the buildings.

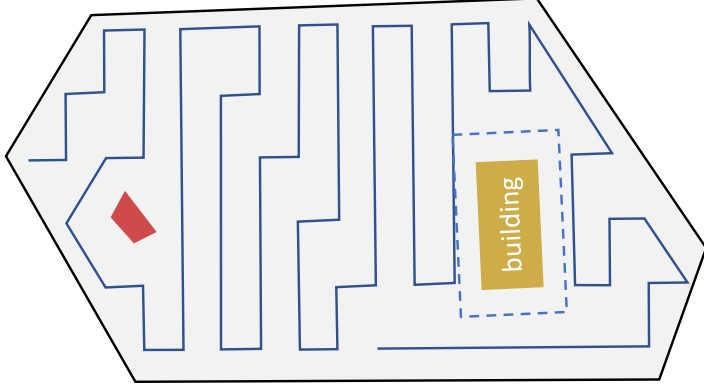

**Figure 29.** Complete coverage path with forbidden flight zone and building generated by GA.

The approach proposed by Trujillo et al. [76] is extended by Darrah et al. [97] for coverage missions over larger areas using multiple multi-rotors. The area of interest is partitioned into equitable sub-areas to be covered by multiple vehicles or by several flights performed by a single vehicle. The partitioning method applies the flood fill algorithm integrated with game theory. In this scenario, each UAV is a player and has a starting position. These players take turns flooding the neighbor cells according to a predefined pattern in a diamond shape, as illustrated in Figure 30. The corresponding order is from left to right, flooding the closest neighbors. The players can not flood building cells or cells previously occupied by other players. The sub-areas are not equally sized. A smaller sub-area may contain building structures to be mapped, while a larger area may present areas of avoidance. Thus, the partitioning method balances the tasks and guarantees an approximate amount of work for each assigned UAV. The coverage paths for each sub-area are obtained using an improved version of the approach proposed by Trujillo et al. [76]. The changes ensure that the multi-rotor's path ends as close as possible to where it started.

| 45 | 39 | 29 | 18 | 31 | 41 | 47 |
|----|----|----|----|----|----|----|
| 37 | 27 | 16 | 8 | 20 | 33 | 43 |
| 25 | 14 | 6 | 2 | 10 | 22 | 35 |
| 13 | 5 | 1 | S | 4 | 12 | 24 |
| 26 | 15 | 7 | 3 | 11 | 23 | 36 |
| 38 | 28 | 17 | 9 | 21 | 34 | 44 |
| 46 | 40 | 30 | 19 | 32 | 42 | 48 |

**Figure 30.** Flood fill pattern with the starting position S and the ordered neighbor cells to be flooded from dark blue to light blue.

Hayat et al. [98] proposed a Multi-Objective Path Planning (MOPP) with a Genetic Algorithm for Search and Rescue (SAR) missions using multiple UAVs. The mission is composed by two steps, search and response. The former one monitors an event (e.g., stationary target) by guaranteeing the whole coverage in a given area. The latter one spreads detection updates on the network. The planning task that occurs during the search phase is performed in a centralized way by the MOPP algorithm, while the time-to-complete the mission is minimized by the GA. This time comprises the period to discover the target and the period to configure a communication trajectory. Thus, the approach needs to optimize two main features, the area coverage and the network connectivity.

5.2.7. Ant Colony Optimization

Ant Colony Optimization (ACO) is adapted for coverage with multiple UAVs by Kuiper and Nadjm-Tehrani [78]. The vehicles share a virtual pheromone map indicating recently visited areas through high rates of pheromone. These pheromones are repulsive and guide the vehicles to unexplored areas. Based on this study, Rosalie et al. [79] introduce the Chaotic Ant Colony Optimization to Coverage (CACOC), which is a technique integrating the ACO with the chaotic dynamical system to surveillance missions in a military context. The approach allows the GCS operator to forecast the UAVs paths, while keeping it unpredictable to the enemies. Although there is no need for communication among vehicles and the base to track the position of the vehicles, the swarm of vehicles still shares a virtual pheromone map. Rosalie et al. [99] further explore the CACOC approach by evaluating its coverage performance by using the V-Rep simulation environment [100].

Cheng et al. [77] propose another bio-inspired approach for cooperative coverage. This approach represents the path of each vehicle as the B-spline curve containing control points, as illustrated in Figure 31a. This optimization problem consists of maximizing the desirability of a path combining four functions: (i) path length, (ii) minimum turning angle, (iii) maximum pitch rate, and (iv) the superposition of the actual trajectory over different UAVs trajectories. The authors consider that the vehicle always moves from left to right, so the first and the last control points are at the borders of the area. The ACO algorithm optimizes the y-axis in the intermediate control points to maximize the coverage. During the algorithm iterations, several ants are launched in the scenario, passing by the initial, intermediate and final points.

Gaussian distribution functions represent the pheromone concentration left by each ant at the control points, being superposed to create a joint distribution function, illustrated in Figure 31b. The resulting function is rescaled to generate the probability density function, which indicates the amount of pheromones in different y-positions. During subsequent iterations, the ants select the control point positions following this probability density function. This function has a pheromone evaporation factor to avoid local optimal points and it is updated at each iteration. In the end of the

algorithm, the zones comprising the major amount of pheromones are selected as intermediate control points to create a complete path for the UAV.

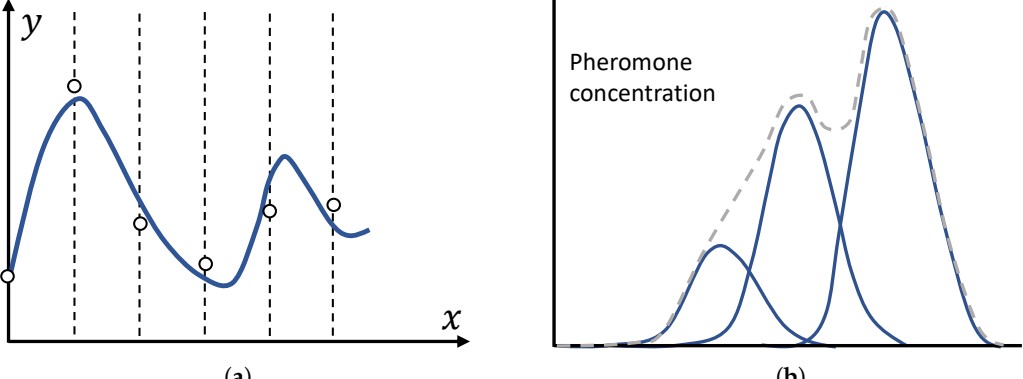

**Figure 31.** Cooperative approach using ACO with Gaussian distribution functions: (**a**) Control points; (**b**) Gaussian distribution functions.

## 6. Discussion

The coverage path planning problem with UAVs has been addressed by several authors in the literature, exploring areas of interest with different shapes and complexities. Generally, simple areas of interest, such as rectangular and convex polygons, do not require any discretization method or decomposition technique, being explored by back-and-forth and spiral flight patterns. The back-and-forth pattern usually defines the sweep direction based on the major axis and uses 90° angles for the turning maneuvers. In polygonal cases, the turning angle may vary both back-and-forth and spiral. In general, these patters require low computational time to find coverage paths and are easy to perform by the aerial platform. One of the main issues involving these platforms is the quite limited flight endurance to perform coverage missions.

Several studies seek to minimize distance, flight time or maneuvers in order to decrease energy consumption. During turning maneuvers, the vehicles should decelerate, rotate and accelerate, extending flight time and, consequently, energy consumption. An alternative is smoothing turning maneuvers to keep the velocity of the vehicles constant, as proposed by Artemenko et al. [31]. Recent efforts developed state-of-the-art energy-aware back-and-forth [25] and spiral [40] algorithms mainly concerned about energy consumption considering resolution and energy constraints of the UAVs. These approaches adopt an energy model explored for analyzing the energetic behavior in distinct circumstances. Thus, optimizing velocity in straight parts of the path leads to energy consumption minimization. Table 1 summarizes the CPP approaches revised in this paper considering areas of interest with no cellular decomposition technique. The table presents the CPP approach, corresponding reference, shape of the area of interest, adopted performance metrics to evaluate coverage pattern, indication of single or multiple UAVs used in the coverage, and the type of UAV: rotary-wing (RW), fixed-wing (FW), or both.

In larger and more complex areas of interest, an exact cellular decomposition may be applied to split the scenario into subregions. The resulting sub-regions can be covered by different sweep directions to obtain an optimal coverage [43,44]. In such areas, there is the possibility to explore four back-and-forth alternatives, varying the direction and the orientation, aiming at minimizing the distance among the subareas, as presented by Torres et al. [35]. Cooperative strategies are also being explored using area decomposition according to the relative capabilities of the vehicles [34]. Obstacles and forbidden flight zones may also be considered in the scenarios [30,48] and, depending on the mission, it might be necessary to revisit previously explored areas [56,60]. A hybrid decomposition technique mixing exact and approximate cellular decomposition is proposed to discretize the area of interest into triangle cells matching almost the exact shape of the area [51,52,54,55]. The authors employ a spiral-like pattern to perform the coverage in a complex area. A decentralized approach is employed for partitioning and

area coverage in regular and irregular-shaped scenarios using heterogeneous aerial vehicles [57–59,101]. Table 2 summarizes the CPP approaches revised in this paper considering areas of interest discretized through the exact cellular decomposition technique. The online/offline column in the table refers to how the coverage path is obtained. In offline cases, the whole path is computed before being performed, while in online cases the path can be computed or modified during coverage.

Depending on the complexity of the area of interest, simple flight patterns may generate inefficient trajectories, requiring post-processing phases to adjust the waypoint positions and the angle between them to overcome this issue. A considerable computational time may be spent evaluating all different rotations to find the optimal sweep direction. Regarding spiral motions, shorter paths can be generated in polygonal shapes with larger internal angles, but in more complex forms, the algorithm may be stuck or may not complete the coverage task. Thus, more sophisticated approaches based on approximate cellular decomposition have been proposed for coverage with UAVs. In Section 4, we decided to separate the existing approaches according to the available information to compute and perform the coverage. When the UAV has full knowledge about the workspace, including area and NFZ, the path can be computed in an offline phase before its execution. However, in some cases, the aerial vehicle does not have complete information about the map due to the presence of moving targets, obstacles, or other vehicles. In this case, the UAV should gather the necessary information using its sensors to perform the mission, usually interleaving between planning and execution.

Considering full information, grid-based solutions exploring the Wavefront algorithm for single UAV are presented by Valente et al. [32] and further optimized using cubic interpolation by Nam et al. [62]. Cooperative strategies using multiple vehicles are introduced by Barrientos et al. [10], including area subdivision and vehicle allocation. A meta-heuristic algorithm named Harmony Search (HS) is proposed by Valente et al. [64] to minimize the number of turning maneuvers in irregular areas and it is compared with the approach proposed by Barrientos et al. [10]. Parts of the area of interest may be covered with distinct resolutions through flights at different altitudes, according to the importance of each subregion [65,66] or depending on the different footprint sensors onboard the vehicles [67].

Cooperative bio-inspired techniques have been dealing with environments containing only partial information. Real-time search methods are considered in simulations and a Node Counting algorithm is applied in real flights by Nattero et al. [36]. A reinforced random walk strategy is introduced by Albani et al. [71] for field coverage and weed mapping, where the UAVs prefer to explore the areas ahead of its current position aligned with the momentum vector. A cellular automata approach is proposed by Zelenka and Kasanicky [73] and Zelenka and Kasanicky [74], but presents problems regarding the pheromone degradation and lack of evaporation. Degradation may generate communication failures between the vehicles, while the excess of pheromones may block certain locations of the area for the vehicles. Some approaches consider the local uncertainty during the coverage missions [81], while other ones explore distributed information merging, where the UAVs directly exchange information to speed up the search and enhance the performance [80]. Path replanning in a continuous UAV trajectory is also explored, where the authors fuse visual information received into a single probabilistic map [82,83].

Another continuous coverage approach using a learning algorithm is addressed by Ramasamy and Ghose [84], considering the preferential surveillance problem. In this problem, the aerial vehicle should increase the visitation frequency in regions of interest and reduce the visitation frequency in a risky region. GA combined with pheromone strategies are also explored [75,76,97,98], while ACO using pheromone maps are introduced for coverage missions with UAVs [77,79,99]. Table 3 summarizes the CPP approaches revised in this paper considering areas of interest discretized through the approximate cellular decomposition technique considering full and partial information. While the CPP approaches with no decomposition or combined with exact cellular decomposition can be executed by rotary-wing, fixed-wing, or both types of vehicles, the CPP approaches using approximate cellular decomposition almost exclusively adopt rotary-wing UAVs. This is because the rotary-wings present maneuverability advantages when making turns in scenarios discretized into a grid. The fixed-wing has maneuverability restrictions, demanding a large space to make turns.

**Table 1.** Coverage path planning approaches in areas of interest with no decomposition technique

| Approach | Ref. | Shape of the area | Performance metrics | Single/ Multiple | Type |
|---|---|---|---|---|---|
| Back-and-Forth, Square, Sector Search, Barrier Patrol | [38] | Rectangular | Fixed and mobile target detection; Coverage rate | Single | RW |
| Back-and-Forth | [39] | Polygonal | Flight time | Single | FW |
| Energy-aware Back-and-Forth | [25] | Polygonal | Energy consumption | Single | RW |
| Energy-aware Spiral | [40] | Polygonal | Energy consumption | Single | RW |
| Three-stage Energy-aware | [33] | 3D Topology | Energy consumption | Single | RW |
| Smoothing algorithms: E-MoTA e I-MoTA | [31] | Regular Grid | Energy consumption; Mission time; Level of localization accuracy | Single | Both |
| Mixed Integer Linear Programming (MILP) | [41] | Rectangular | Flight time | Multiple | FW |
| Circular | [42] | Rectangular | Coverage rate; time | Multiple | FW |

**Table 2.** Coverage path planning approaches in areas of interest with exact cellular decomposition

| Approach | Ref. | Online/ Offline | Shape of the area | Performance metrics | Single/ Multiple | Type |
|---|---|---|---|---|---|---|
| Back-and-Forth | [43,44] | Offline | Polygonal | Number of turning maneuvers | Single | Both |
| Back-and-Forth | [35] | Offline | Polygonal | Number of turning maneuvers; Path length | Single | RW |
| Back-and-Forth | [30,48] | Offline/ Online | Irregular | Path length; Coverage time | Single | FW |
| Back-and-Forth and Spiral | [29] | Offline | Polygonal | Path length | Single | FW |
| Back-and-Forth | [34] | Offline | Polygonal | Number of turning maneuvers | Multiple | RW |
| Spiral | [51,52,54,55] | Offline | Polygonal | Path length | Multiple | FW |
| Back-and-Forth (Line Formation) | [56] | Offline | Rectangular | Target detection; Search time, number of UAVs and information exchange | Multiple | RW |
| One-to-one coordination (Decentralized Technique) | [57–59] | Online | Irregular | Interval of visits; Information latency | Multiple | Both |
| Back-and-Forth/Zamboni (Local Priority) | [60] | Offline | Polygonal | Number of turning maneuvers; uncertainty | Multiple | FW |

**Table 3.** Coverage path planning approaches in areas of interest with approximate cellular decomposition

| Approach | Ref. | Online/Offline | Shape of the area | Performance metrics | Single/Multiple | Type |
|---|---|---|---|---|---|---|
| Gradient-based approach | [32] | Offline | Irregular/Regular Grid | Coverage time | Single | RW |
| Wavefront Algorithm and Cubic Interpolation | [62] | Offline | Irregular/Regular Grid | Path length; Number of turning maneuvers | Single | RW |
| Multi-RTT* Fixed Node (RRT*FN) and Genetic Algorithm (GA) | [63] | Offline | Regular Grid | Path length | Single | RW |
| Wavefront Algorithm | [10] | Offline | Irregular/Regular Grid | Position and altitude errors; Wind disturbances; Mission, flight and configuration times; Path length | Multiple | RW |
| Harmony Search | [64] | Offline | Irregular/Regular Grid | Number of turning maneuvers | Multiple | RW |
| Breadth-First strategy, Depth-First, and Shortcut Heuristic | [65] | Online | Square | Total distance of coverage | Single | RW |
| Hilbert space-filling curves | [66] | Online | Square | Total distance of coverage | Single | RW |
| Long straight-lines algorithm | [67] | Offline | Irregular/Grid related to the sensor | Total distance; Number of turns; Number of jumps between cells | Multiple | RW |
| Edge Counting and PatrolGRAPH* | [36] | Online | Graph Grid | Path length; Robots distance average | Multiple | RW |
| Reinforced Random Walk | [71] | Online | Rectangular | Coverage time; Global detection efficiency | Multiple | RW |
| Cellular Automata | [72–74] | Online | Regular Grid | Exploration time with/without barriers | Multiple | RW |
| Waypoint planning with uncertainty | [81] | Online | Rectangular | Certainty of information points | Multiple | Both |
| Information merging for cooperative search | [80] | Online | Rectangular | Target localization | Multiple | RW |
| Fixed-horizon with CMA-ES | [82,83] | Online | Rectangular | Entropy; Classification rate | Multiple | RW |
| Learning-based Preferential Surveillance Algorithm (LPSA) | [84,85] | Online | Regular Grid | Distribution of visits; Target localization; Threat avoidance | Single | Both |
| Back-and-Forth | [75] | Online | Regular Grid | Total distance; Coverage rate; Redundancy rate | Multiple | RW |
| Genetic Algorithm (GA) | [76] | Offline/Online | Polygonal/Regular Grid | Path length | Single | RW |
| GA with flood fill algorithm | [97] | Offline/Online | Polygonal/Regular Grid | Path length | Multiple | RW |
| Multi-Objective Path Planning with GA | [98] | Offline/Online | Rectangular | Mission Completion Time | Multiple | RW |
| Chaotic Ant Colony Optimization to Coverage | [79,99] | Offline/Online | Regular Grid | Coverage rate; Recent coverage ratio; Fairness (coverage distribution); Connectivity (UAVs distribution) | Multiple | RW |
| ACO with Gaussian distribution functions | [77] | Online | 3D Regular Grid | Path length and rotation angle; Inclination and area overlapping rate | Multiple | Both |

The algorithms proposed to solve the CPP problem are usually concerned with the planning phase to obtain a coverage path according to a performance metric. When dealing with fixed-wing UAVs, the approaches must also consider motion constraints of such vehicles in order to plan a feasible trajectory. However, these approaches do not consider important control tasks such as path following or trajectory tracking, trusting exclusively the internal controller of the UAV to perform the planned path in real flights.

## 7. Conclusions

This paper presents a survey on coverage path planning with unmanned aerial vehicles, addressing simple geometric flight patterns, such as back-and-forth and spiral, and more complex grid-based solutions considering full and partial information about the area of interest. The surveyed coverage approaches are classified according to the classical taxonomy defined by Choset [9]; no decomposition, exact and approximate cellular decomposition. The study contemplates different shapes of the area of interest, such as rectangular, concave and convex polygons. We also present performance metrics usually applied to evaluate the success of the coverage missions.

The limited endurance of UAVs is the major concern to be overcome to perform more complex coverage missions. Some authors employ multiple vehicles to enhance the coverage performance in such missions, splitting long and high energy demand flights in more feasible flights. However, this technique usually demands computational complexity to solve coordination and communication issues. This cooperative context still lacks a more robust solution to handle problems more autonomously. The area of interest is usually discretized, divided and assigned to the members of a team using a centralized decision-making process. Moreover, communication between vehicles and a ground control station is required for coordination, which is not robust to failures or feasible in real-world scenarios.

The proposed approaches aim to minimize the path length, the mission execution time, and the number of turning maneuvers to indirectly save energy. However, in some cases, performance metrics such as path length and energy consumption may be conflicting, once shorter paths may contain more abrupt maneuvers consuming more energy. As stated by Di Franco and Buttazzo [24] and Cabreira et al. [40] another issues, such as vehicle dynamics, turning angle, and optimal speed should be considered to minimize energy consumption. But these energy-aware approaches are still restricted to regular scenarios considering only simple flight patterns.

We recently proposed a state-of-the-art energy-aware spiral coverage algorithm for regular areas and we are currently going forward to a global solution considering more complex scenarios with a grid-based method. Furthermore, we are also developing a decentralized approach for cooperative coverage path planning with UAVs considering communication and energy constraints. In the future, we intend to develop an efficient synchronization mechanism among the vehicles to avoid the need for a ground control station. Finally, we are studying the impact of the vehicle dynamics and the external environment conditions in a physics-based energy model for the energy-aware mission planning. With our current survey and future works in coverage path planning problems, we expect to advance the state-of-the-art in real-world feasible mission planning approaches for autonomous aerial vehicles.

**Author Contributions:** Conceptualization, T.M.C., L.B.B., and P.R.F.J.; Methodology, T.M.C., L.B.B., and P.R.F.J.; Investigation, T.M.C.; Writing-original draft preparation, T.M.C.; Writing-review and editing, T.M.C., L.B.B., and P.R.F.J.; Supervision, L.B.B. and P.R.F.J.; project administration, P.R.F.J.; funding acquisition, L.B.B. and P.R.F.J.

**Funding:** This research was funded in part by CAPES - Finance Code 001, by FAPERGS/CNPq grant number 16/2551-0000/472-2 and by CNPq grant number 308487/2017-6.

**Conflicts of Interest:** The authors declare no conflict of interest. The funders had no role in the design of the study; in the collection, analyses, or interpretation of data; in the writing of the manuscript, or in the decision to publish the results.

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
