# Peer review of "Survey on Coverage Path Planning with Unmanned Aerial Vehicles"

_drones, doi:10.3390/drones3010004_

Round 1
Reviewer 1 Report
This paper is a good comprehensive review of CPP for UAVs, cover all major areas and topics.
There are quite a lot of grammatical errors which should be addressed, as the ruins the flow of the text e.g
Page1 line 19 "The UAVs comprise aerial platforms without the presence of pilots"
Pages 2 line 75 "it is presented the classical survey on CPP"
There are many more
In the motivation section you mention ground vehicle CPP, what are the differences between CPP for ground vehicle compared to UAVs
Page 6 an explanation of GSD is needed as this is a key parameter when planning a remote sensing UAV survey
As you mention BF path a lot it would be good to mention that they are the most efficient CPP paths for a convex polygon as proved in Li 2011
Your exact cellular decomposition section is huge and the papers you review don't seem to be in any order. Add subsections and group the papers accordingly
Most papers were reviewed however there are a couple that the author should include.
On exact cellular decomposition: Optimal polygon decomposition for UAV survey coverage path planning in wind Matthew Coombes, Tom Fletcher, Wen-Hua Chen, Cunjia Liu (which is an extension of ref [39])
Rotary wing a fixed wing have been lumped together in this paper perhaps a discussion on which techniques are better for which.
All these CPP techniques require excellent path following performance (especially hard for fixed wing) from the UAV perhaps some mention of techniques that account for this. Or at least a discussion on the importance of this.
Author Response
This paper is a good comprehensive review of CPP for UAVs, cover all major areas and topics.
There are quite a lot of grammatical errors which should be addressed, as the ruins the flow of the text e.g
Page1 line 19 "The UAVs comprise aerial platforms without the presence of pilots"
Pages 2 line 75 "it is presented the classical survey on CPP"
There are many more.
Authors: We reviewed the text correcting the grammatical errors.
In the motivation section you mention ground vehicle CPP, what are the differences between CPP for ground vehicle compared to UAVs
Authors: The CPP approaches for ground and aerial vehicles are similar, but there are more restrictions that should be considered when dealing with UAVs, as stated below:
“Although land exploration techniques revised in the previous surveys can be extended and applied to UAVs, several additional aspects must be considered when dealing with aerial vehicles such as vehicle's physical characteristics, endurance, maneuverability limitations, restricted payload, environmental external conditions, among others.”
In general, the approaches for aerial vehicles should concern about the energy consumption during the mission, once such vehicles have limited endurance. In this way, the approaches usually try to minimize the flight time or the number of turns in order to save energy.
Page 6 an explanation of GSD is needed as this is a key parameter when planning a remote sensing UAV survey
Authors: We thanks the reviewer for pointing out the lack of clarity in this portion of the manuscript. We added an explanation about GSD (line 205):
“The GSD is the length on the ground corresponding to the side of one pixel in the image, or the distance between pixel centers measured on the ground. The lower the flight altitude of the UAV, smaller the GSD and the better the image quality.”
As you mention BF path a lot it would be good to mention that they are the most efficient CPP paths for a convex polygon as proved in Li 2011
Authors: In fact, the BF is explored by several authors in literature and it is the most popular CPP approach. However, we shown in our paper [40] that the E-Spiral technique is the most efficient CPP approach for convex polygons considering the real energy spent. We edited the manuscript with this affirmation when we mention the ref. [40] in lines 331-332:
Cabreira, T.M.; Di Franco, C.; Ferreira Jr., P.R.; Buttazzo, G.C. Energy-Aware Spiral Coverage Path Planning for UAV Photogrammetric Applications. IEEE Robotics and Automation Letters 2018, 3, 3662–3668. doi:10.1109/LRA.2018.2854967.
Your exact cellular decomposition section is huge and the papers you review don't seem to be in any order. Add subsections and group the papers accordingly
Authors: We followed the reviewer’s suggestions and added subsections splitting the papers into single and cooperative strategies. The cooperative strategies are further split into BF, spiral, line formation, decentralized technique, and local priority.
Most papers were reviewed however there are a couple that the author should include on exact cellular decomposition: Optimal polygon decomposition for UAV survey coverage path planning in wind Matthew Coombes, Tom Fletcher, Wen-Hua Chen, Cunjia Liu (which is an extension of ref [39])
Authors: We thanks the reviewer for this contribution. We included the referred paper in the exact cellular decomposition section (page 13).
Rotary wing a fixed wing have been lumped together in this paper perhaps a discussion on which techniques are better for which.
Authors: We added a column in tables 1-3 with the types of UAVs (rotary-wing, fixed-wing or both) adopted in each approach, as requested by reviewer #3. We also added a discussion about the suitable types of vehicles for the approaches at the end of Section 7:
“While the CPP approaches with no decomposition or combined with exact cellular decomposition can be executed by rotary-wing, fixed-wing, or both types of vehicles, the CPP approaches using approximate cellular decomposition adopt almost exclusively rotary-wing UAVs. This is because the rotary-wings present maneuverability advantages when making turns in scenarios discretized into a grid. The fixed-wing has maneuverability restrictions, demanding a large space to make turns.”
All these CPP techniques require excellent path following performance (especially hard for fixed wing) from the UAV perhaps some mention of techniques that account for this. Or at least a discussion on the importance of this.
Authors: The reviewer is right saying that the path following is important. However, this issue is usually addressed in control papers in the literature. We added a paragraph at the end of Section 7 stating the importance of this issue:
“The algorithms proposed to solve the CPP problem usually concern about the planning phase to obtain a coverage path according to a performance metric. When dealing with fixed-wing UAVs, the approaches must also consider the motion constraints of such vehicles in order to plan a feasible trajectory. However, these approaches do not consider important control tasks such as path following or trajectory tracking, trusting exclusively in the UAV internal controller to perform the planned path in real flights.”

Reviewer 2 Report
The manuscript presents a comprehensive survey about the coverage path planning algorithms proposed for unmanned aerial vehicles(UAVs) in recent years. The survey covers the simple geometric flight patterns, area decomposition techniques, different types of the area of interest used for coverage missions, performance metrics used for coverage path planning algorithms evaluation, coverage types, information availability about the area of interest and path smoothing mechanisms for single and multiple UAVs. Additionally, the authors classified and summarized the several coverage path planning algorithms proposed in recent years based on the types of decomposition.
The followings are not musts, but the authors may consider to include or add to improve the paper.
- path planning for spatially distributed regions coverage
- computational complexities were not dealt with
- more path optimization algorithms
- time optimization aspects in addition to the energy optimization
Miner Comments
The overall paper is well-written and clear. Meanwhile, I would suggest authors to verify or correct following English errors.
On page #: 3, Line #: 117, the following sentence needs correction
This surveys has the following structure
On page #: 6, Line #: 212, the spelling of travelled needs correction/verification in following sentence.
in the literature are: the total traveled distance or the path length
On page #: 8, Line #: 284, The author can verify the use of verb in the following sentence, I think continuing is proper word instead of continue.
From that, instead of continue the circular trajectory, it…
On page #: 12, Line #: 372, the use of the ‘an’ seems improper in the following sentence.
one subregion to another in order to obtain an coverage improvement,
On page #: 16, Line #: 491, Following sentence also needs correction.
However, sometimes this pattern do not complete the coverage in difficult areas.
On page #: 17, Line #: 520, the following sentence need correction regarding the correct use of the verb.
In both cases, the approximate cellular decomposition can be employed to discretized the area of interest into a grid,
On page #: 26 and 27, Line #: 785, 826, Incorrect use of the verb forms in the following two sentences.
The former one monitor an event (e.g., stationary target) by guaranteeing the whole coverage in a given area.
In general, these patterns demands low computational time to be searched and are easy to perform by the aerial platform.
The acronyms should be carefully defined the first time they are used. Meanwhile, the abbreviation DLS on page #: 17, line #: 537, has not been defined for the first time.
Author Response
The manuscript presents a comprehensive survey about the coverage path planning algorithms proposed for unmanned aerial vehicles (UAVs) in recent years. The survey covers the simple geometric flight patterns, area decomposition techniques, different types of the area of interest used for coverage missions, performance metrics used for coverage path planning algorithms evaluation, coverage types, information availability about the area of interest and path smoothing mechanisms for single and multiple UAVs. Additionally, the authors classified and summarized the several coverage path planning algorithms proposed in recent years based on the types of decomposition.
The followings are not musts, but the authors may consider to include or add to improve the paper.
- path planning for spatially distributed regions coverage
- computational complexities were not dealt with
- more path optimization algorithms
- time optimization aspects in addition to the energy optimization
Authors: We thanks the reviewer for the suggestions. We included the paper “Optimal polygon decomposition for UAV survey coverage path planning in wind Matthew Coombes, Tom Fletcher, Wen-Hua Chen, Cunjia Liu (which is an extension of ref [39])” in the exact cellular decomposition section (page 13). This paper deals with path optimization exploring different polygon rotations and recombination of cells. The authors also proposed a cost function for time optimization in the presence of wind, comparing the proposed approach with previous techniques that minimize the number of turns and the sum of altitudes.
The overall paper is well-written and clear. Meanwhile, I would suggest authors to verify or correct following English errors.
On page #: 3, Line #: 117, the following sentence needs correction
This surveys has the following structure
On page #: 6, Line #: 212, the spelling of travelled needs correction/verification in following sentence.
in the literature are: the total traveled distance or the path length
On page #: 8, Line #: 284, The author can verify the use of verb in the following sentence, I think continuing is proper word instead of continue.
From that, instead of continue the circular trajectory, it…
On page #: 12, Line #: 372, the use of the ‘an’ seems improper in the following sentence.
one subregion to another in order to obtain an coverage improvement,
On page #: 16, Line #: 491, Following sentence also needs correction.
However, sometimes this pattern do not complete the coverage in difficult areas.
On page #: 17, Line #: 520, the following sentence need correction regarding the correct use of the verb.
In both cases, the approximate cellular decomposition can be employed to discretized the area of interest into a grid,
On page #: 26 and 27, Line #: 785, 826, Incorrect use of the verb forms in the following two sentences.
The former one monitor an event (e.g., stationary target) by guaranteeing the whole coverage in a given area.
In general, these patterns demands low computational time to be searched and are easy to perform by the aerial platform.
Authors: We reviewed the text correcting the grammatical errors.
The acronyms should be carefully defined the first time they are used. Meanwhile, the abbreviation DLS on page #: 17, line #: 537, has not been defined for the first time.
Authors: The abbreviation was defined for the first time.

Reviewer 3 Report
The paper conducts an extensive survey into Coverage Path Planning (CPP) techniques that focus on Unmanned Aerial Vehicles (UAVs). In a nutshell, the CPP deals with finding a trajectory, or multiple trajectories (if multiple UAVs are involved), in order to fully explore a given area. The area might be of a different shape (e.g. rectangular, convex, concave) and, moreover, only partial information about the area might be available. The trajectories have to comply with dynamic and energy constraints of the given UAVs with the aim of being energy efficient (as it is usually used as optimization criterion).
The paper delivers what it promises and what one would expect from a survey paper. The state-of-the-art techniques addressing the CPP problem on UAVs are appropriately categorized, according to the shape of the area to cover, full/partial observability and single/multiple UAV support to mention some. The paper does not go into very details when describing the techniques, which is understandable (otherwise the paper would have been very long). In a few instances, however, the description of the techniques is unclear - for example, speaking about the work of Truillo et al. [75] (page 25) it is stated that "The partitioning method applies the flood fill algorithm integrated within the game theory". Such an information is rather confusing (at least I don't have a clue how the method might work) and can either be explained in more detail, committed, or some more references should be given.
The high-level overview of the discussed techniques as is done in tables 1-3 makes it much easier for the reader to "pick up" techniques that are relevant for a class of CPP problems s/he is interested. If possible, the list of categories could be expanded by considering whether a given technique is suitable for fixed-wing or rotary-wing UAVs (or both), as the differences are thoroughly discussed in the introductory section, and whether a given technique reasons with No Fly Zones.
Whereas addressing the above issues is "nice to have", the paper has a lot of grammar issues that has to be addressed prior publication (the following list is not exhaustive - I recommend the authors to proofread the paper):
- citations do not stand for nouns. Constructs such as "In [9], it is presented .." should be rewritten to "Choset [9] presents ...", or "... presented in [20]" should be rewritten to "... presented by Gautam et al. [20]".
- page 1 line 5: "the existent studies" -> "the existing studies"
- page 4 line 133: "environment usually is split" -> "environment is usually split"
- page 4 line 153: "the UAV are not allowed to fly" - > "the UAVs are not allowed to fly" (or "the UAV is not allowed to fly")
- page 4 line 157: "concerns about the CPP problem is guarantee" - > "concerns about the CPP problem is to guarantee"
- page 6 line 198: "the possibility of employ" -> "the possibility to employ"
- page 7 line 224: "the relatives capabilities" - > "the relative capabilities"
- page 7 line 232: "increase again" -> "increase its speed again" ?
- page 8 line 282: "consists in" -> "consist of" (the issue can be found multiple times in the paper)
- page 9 line 299: "that is possible" -> "that it is possible"
- page 9 line 304: "cover previous explored zones" -> "cover previously explored zones"
- page 10 line 326: "The first step is build" -> "The first step is to build"
- page 21 line 631: "biological-inspired" - > "biologically-inspired"
- page 24 line 748: "pass for selection" -> "pass through the selection"
- page 26 line 798: "despite there is no need" -> "Although there is no need"
- page 27 line 826: "these patters demands low computational time to be searched" -> "these patters require low computational time to find" ?
Author Response
The paper conducts an extensive survey into Coverage Path Planning (CPP) techniques that focus on Unmanned Aerial Vehicles (UAVs). In a nutshell, the CPP deals with finding a trajectory, or multiple trajectories (if multiple UAVs are involved), in order to fully explore a given area. The area might be of a different shape (e.g. rectangular, convex, concave) and, moreover, only partial information about the area might be available. The trajectories have to comply with dynamic and energy constraints of the given UAVs with the aim of being energy efficient (as it is usually used as optimization criterion).
The paper delivers what it promises and what one would expect from a survey paper. The state-of-the-art techniques addressing the CPP problem on UAVs are appropriately categorized, according to the shape of the area to cover, full/partial observability and single/multiple UAV support to mention some. The paper does not go into very details when describing the techniques, which is understandable (otherwise the paper would have been very long). In a few instances, however, the description of the techniques is unclear - for example, speaking about the work of Truillo et al. [75] (page 25) it is stated that "The partitioning method applies the flood fill algorithm integrated within the game theory". Such an information is rather confusing (at least I don't have a clue how the method might work) and can either be explained in more detail, committed, or some more references should be given.
Authors: We thanks the reviewer for pointing out the lack of clarity in this portion of the manuscript. We explained how the flooding method works (line 841) and added a figure to help the understanding (Figure 30).
The high-level overview of the discussed techniques as is done in tables 1-3 makes it much easier for the reader to "pick up" techniques that are relevant for a class of CPP problems s/he is interested. If possible, the list of categories could be expanded by considering whether a given technique is suitable for fixed-wing or rotary-wing UAVs (or both), as the differences are thoroughly discussed in the introductory section, and whether a given technique reasons with No Fly Zones.
Authors: We followed the reviewer’s suggestions and added a column in tables 1-3 with the types of UAVs adopted in each approach. We also added a discussion about the suitable types of vehicles for the approaches at the end of Section 7:
“While the CPP approaches with no decomposition or combined with exact cellular decomposition can be executed by rotary-wing, fixed-wing, or both types of vehicles, the CPP approaches using approximate cellular decomposition adopt almost exclusively rotary-wing UAVs. This is because the rotary-wings present maneuverability advantages when making turns in scenarios discretized into a grid. The fixed-wing has maneuverability restrictions, demanding a large space to make turns.”
Whereas addressing the above issues is "nice to have", the paper has a lot of grammar issues that has to be addressed prior publication (the following list is not exhaustive - I recommend the authors to proofread the paper):
Authors: We reviewed the text correcting the grammatical errors.
- citations do not stand for nouns. Constructs such as "In [9], it is presented .." should be rewritten to "Choset [9] presents ...", or "... presented in [20]" should be rewritten to "... presented by Gautam et al. [20]".
Authors: We thanks the reviewer for pointing out these issues. All constructs were rewritten along the manuscript.
- page 1 line 5: "the existent studies" -> "the existing studies"
- page 4 line 133: "environment usually is split" -> "environment is usually split"
- page 4 line 153: "the UAV are not allowed to fly" - > "the UAVs are not allowed to fly" (or "the UAV is not allowed to fly")
- page 4 line 157: "concerns about the CPP problem is guarantee" - > "concerns about the CPP problem is to guarantee"
- page 6 line 198: "the possibility of employ" -> "the possibility to employ"
- page 7 line 224: "the relatives capabilities" - > "the relative capabilities"
- page 7 line 232: "increase again" -> "increase its speed again" ?
- page 8 line 282: "consists in" -> "consist of" (the issue can be found multiple times in the paper)
- page 9 line 299: "that is possible" -> "that it is possible"
- page 9 line 304: "cover previous explored zones" -> "cover previously explored zones"
- page 10 line 326: "The first step is build" -> "The first step is to build"
- page 21 line 631: "biological-inspired" - > "biologically-inspired"
- page 24 line 748: "pass for selection" -> "pass through the selection"
- page 26 line 798: "despite there is no need" -> "Although there is no need"
- page 27 line 826: "these patters demands low computational time to be searched" -> "these patters require low computational time to find" ?
Authors: We reviewed the text correcting the grammatical errors.
